# Allosteric regulation of glycogen breakdown by the second messenger cyclic di-GMP

**Maria A. Schumacher** [1,7] ✉, **Mirka E. Wörmann**[2,6,7], **Max Henderson**[1],
**Raul Salinas**[1], **Andreas Latoscha** [2], **Mahmoud M. Al-Bassam**[3],
**Kumar Siddharth Singh**[4], **Elaine Barclay**[5], **Katrin Gunka**[4] & **Natalia Tschowri** [4] ✉

*Streptomyces* are our principal source of antibiotics, which they generate concomitant with a complex developmental transition from vegetative hyphae to spores. c-di-GMP acts as a linchpin in this transition by binding and regulating the key developmental regulators, BldD and WhiG. Here we show that c-di-GMP also binds the glycogen-debranching-enzyme, GlgX, uncovering a direct link between c-di-GMP and glycogen metabolism in bacteria. Further, we show c-di-GMP binding is required for GlgX activity. We describe structures of apo and c-di-GMP-bound GlgX and, strikingly, their comparison shows c-di-GMP induces long-range conformational changes, reorganizing the catalytic pocket to an active state. Glycogen is an important glucose storage compound that enables animals to cope with starvation and stress. Our in vivo studies reveal the important biological role of GlgX in *Streptomyces* glucose availability control. Overall, we identify a function of c-di-GMP in controlling energy storage metabolism in bacteria, which is widespread in Actinobacteria.

In all organisms, excess glucose is converted to polymeric osmotically non-active storage compounds. Starch is the predominant storage carbohydrate in higher plants[1], while bacteria and vertebrates accumulate glycogen as a reserve polymer[2–4]. Glycogen is a polysaccharide largely consisting of α−1,4-linked glucose subunits with α−1,6-linked glucose at the branching points. In humans, glycogen is mainly stored in the liver and the skeletal muscle. The glycogen stored in the liver serves as a glucose reservoir to the bloodstream during fasting periods and to muscle cells during muscle contraction[2]. Mutations in genes encoding enzymes involved in glycogen metabolism lead to glycogen storage disorders, including nervous system, kidney and cardiac dysfunctions[5].

Bacteria convert excess carbon to glycogen under conditions of reduced growth when a nutrient e.g., a nitrogen source is limited and in the stationary phase[6,7]. Early studies demonstrated that glycogen is associated with prolonged survival, likely providing energy needed for cell maintenance when a carbon source is lacking[8,9]. The storage polysaccharide aids survival under nutrient-poor conditions of the pathogen *Vibrio cholerae* and contributes to its pathogenesis[6]. In cyanobacteria, glycogen is pivotal to the viability and resuscitation of dormant, chlorotic cells that are formed under nitrogen starvation and is involved in dark adaptation[10,11]. Glycogen has also been reported to contribute to stress resistance, such as desiccation tolerance in *Pseudomonas aeruginosa* PAO1 and fast adaptation to hyperosmotic stress in *Corynebacterium glutamicum*[12,13].

The classical pathway for glycogen biosynthesis in bacteria involves three major reactions. The adenosine 5′-diphosphate (ADP)-glucose pyrophosphorylase (AGPase; GlgC) catalyzes the formation of ADP-glucose from α-glucose-1-phosphate and ATP. Glycogen synthase (GS; GlgA) generates linear α-(1→4)-linked glucose chains out of ADP-glucose. The glycogen branching enzyme (GBE; GlgB) produces α-(1→6)-linked glucan branches in the polymer. On the other hand, glycogen degradation requires the glycogen debranching enzyme (GDE; GlgX), which directly catalyzes the hydrolysis of α-(1→6)-linked glucose

[1]Department of Biochemistry, Duke University School of Medicine, Durham, NC 27710, USA. [2]Institute for Biology/Microbiology, Humboldt-Universität zu Berlin, 10115 Berlin, Germany. [3]Department of Pediatrics, University of California, San Diego, La Jolla, CA 92093, USA. [4]Institute of Microbiology, Leibniz Universität Hannover, 30419 Hannover, Germany. [5]Department of Cell and Developmental Biology, John Innes Centre, Norwich NR4 7UH, UK. [6]Present address: Bundesinstitut für Risikobewertung, 12277 Berlin, Germany. [7]These authors contributed equally: Maria A. Schumacher, Mirka E. Wörmann. ✉e-mail: maria.schumacher@duke.edu; tschowri@ifmb.uni-hannover.de

residues and the glycogen phosphorylase (GP; GlgP) to generate glucose-1-phosphate, which feeds into primary metabolism[7]. An alternative route to glycogen depends on the maltosyltransferase GlgE that uses the disaccharide α-maltose-1-phosphate as the building block to extend glucan chains. The *glgE* pathway was discovered in *Mycobacterium tuberculosis*[14] and represents the sole route to glycogen biosynthesis in *Streptomyces venezuelae*, which does not encode *glgA*[15,16].

Nucleotide-based second messengers have been reported to control glycogen metabolism at genetic and posttranslational levels in both humans and bacteria. Krebs and Fischer showed that cyclic adenosine (3′,5′)-monophosphate (cAMP) stimulates glycogen breakdown by activating GP via the protein kinase PKA[17]. In *E. coli*, the cAMP producing adenylate cyclase (Cya) and the cAMP receptor protein (CRP) are required for glycogen gene expression and polysaccharide synthesis[18–20]. Guanosine tetraphosphate and guanosine pentaphosphate (collectively termed (p)ppGpp or the alarmone) produced by the RelA/SpoT family of proteins has also been reported to stimulate glycogen production[21–23]. A study by EydallIn et al.[22] linked the second messenger bis-(3′−5′)-cyclic diguanosine monophosphate (c-di-GMP) to the control of glycogen formation in *E. coli*. Using a gene expression library, the authors showed that a strain overexpressing the c-di-GMP-producing diguanylate cyclase (DGC) DgcP has reduced glycogen levels. On the contrary, *E. coli* strains overexpressing the c-di-GMP-hydrolyzing phosphodiesterases, PdeC or Dos, contain more glycogen[22]. However, the mechanism by which this regulation occurs is unknown. Indeed, in contrast to the defined role of cAMP in the regulation of glycogen breakdown in humans, there is a big gap in our knowledge of how second messengers regulate glycogen metabolism in bacteria and whether any of them have a direct effect on the levels of this storage compound.

c-di-GMP is produced by DGCs from GTP through their catalytic GGDEF domains that are named after key amino acids in their active sites[24]. The dinucleotide is hydrolyzed to the linear 5′-phosphoguanylyl-(3′−5′)-guanosine (pGpG) by EAL-domain containing PDEs or to two molecules of GMP by HD-GYP domains[25–27]. c-di-GMP fulfills a global regulatory role in diverse bacterial functions, including biofilm formation, motility, virulence and cell cycle progression by binding to diverse protein effectors and riboswitches[28,29]. In antibiotic-producing soil bacteria *Streptomyces*, c-di-GMP controls a complex transition from filamentous hyphae to spores[30,31]. The life cycle of these actinomycetes begins with spore germination followed by the growth of a dense network of branched hyphae that form the substrate mycelium. Reproduction is initiated by the rise of aerial hyphae that undergo multiple septation events and differentiate into chains of unigenomic spores[32]. During vegetative growth, the master developmental regulator, BldD, binds a unique tetrameric form of c-di-GMP, which promotes its dimerization and function as a repressor of sporulation genes[33,34]. A second key developmental regulator, the sporulation-specific σ-factor WhiG, is also controlled by c-di-GMP. The anti-σ-factor RsiG sequesters WhiG in an inactive complex which is stabilized by a dimer of c-di-GMP[35]. Interestingly, in *Streptomyces coelicolor*, WhiG has been proposed to affect glycogen deposition and thus to coordinate glycogen metabolism with differentiation[36].

In this study, we identify the GDE of *S. venezuelae*, GlgX, as a target of c-di-GMP thus uncovering a direct link between c-di-GMP and glycogen metabolism in this bacterium. We demonstrate that c-di-GMP binding stimulates GlgX-mediated glycogen breakdown and describe crystal structures of *S. venezuelae* GlgX, which reveal the molecular mechanism by which c-di-GMP activates GlgX. Specifically, these structures show that c-di-GMP binding induces structural changes that are transmitted to the active site, resulting in the reorganization of the catalytic pocket. Structures of GlgX-c-di-GMP bound to acarbose confirmed that c-di-GMP stabilizes the correct substrate binding conformation. Thus, these studies uncover

another direct contribution of c-di-GMP to the *Streptomyces* developmental process.

## Results

### c-di-GMP levels fluctuate during developmental growth of *S. venezuelae*

c-di-GMP signals through BldD and WhiG to control developmental program progression in *Streptomyces*[34,35]. To gain insight into the cellular levels of the dinucleotide during the *S. venezuelae* life cycle, we extracted nucleotides from cells grown for up to 20 h in liquid Maltose-Yeast-extract Malt-extract (MYM) medium and performed liquid chromatography-coupled tandem mass spectrometry for quantification (LC-MS/MS). As confirmed by microscopic inspection (Supplementary Fig. 1a, b), *S. venezuelae* initiates differentiation after ca. 12–14 h of growth when grown in liquid MYM medium, which is mainly marked by fragmentation of vegetative hyphae. After ca. 16–18 h of incubation, spores begin to be microscopically visible. In the vegetative phase, we determined $36.1 \pm 5.2$ pmol c-di-GMP/mg protein after 8 h of growth. The levels drop to $7.2 \pm 3.9$ pmol/mg protein during the transition to sporulation after 14 h incubation. Interestingly, during spore formation, c-di-GMP levels increase again and reach a maximum of $68.0 \pm 11.5$ pmol/mg protein (Fig. 1a).

These data are in line with the model that increased c-di-GMP levels activate BldD as a repressor of sporulation genes and stimulate sequestration of the sporulation-specific sigma factor WhiG by RsiG during vegetative growth. As expected, the levels drop upon initiation of development to inactivate BldD and to release WhiG. However, the rise of c-di-GMP during spore formation suggests that the molecule may fulfill an unknown regulatory function in the late developmental stage.

### The glycogen debranching enzyme GlgX (Vnz29170) is a c-di-GMP effector

Increase of c-di-GMP during sporulation (Fig. 1a) suggested the possibility that an up to now unrecognized c-di-GMP effector protein(s) senses the signaling molecule in the late developmental stage. To identify such a putative c-di-GMP-binding protein(s), we relied on our affinity pull-down screen using a c-di-GMP-capture compound[37] that led to the identification of BldD as a c-di-GMP effector[34].

Interestingly, the putative glycogen debranching enzyme GlgX (Vnz29170; SVEN5898) was repeatedly enriched in this pull-down assay. GlgX is a 79 kDa protein that contains a putative carbohydrate-binding module (CBM_48; PF02922) and alpha amylase, catalytic domain (PF00128). Based on our analysis using Clustal 2.1, GlgX shares 51% and 46% sequence identity with the glycogen debranching enzyme TreX from *Sulfolobus solfataricus*[38] and $GlgX_{EC}$ from *E. coli*, respectively[39] (Supplementary Fig. 2). *S. venezuelae* has a second GlgX protein, Vnz25270 (SVEN5112) that displays 58% identity to Vnz29170, which was not identified in our pull-down assay. To examine the expression of Vnz29170, we generated a C-terminal FLAG-tagged allele under the control of the native promoter from the p3xFLAG vector[40], which integrates into the ΦBT1 attachment site in the *S. venezuelae* chromosome. Using a monoclonal anti-FLAG antibody and western blotting, we detected accumulation of Vnz29170-FLAG with maximum levels during sporulation (Supplementary Fig. 3).

To determine if the interaction between Vnz29170 (herein called GlgX) and c-di-GMP is direct, we purified an N-terminally $His_6$-tagged GlgX protein and performed nano differential scanning fluorimetry (nanoDSF) thermal shift assays. We found that in the presence of c-di-GMP, the melting point of $His_6$-GlgX was shifted from 51.5 °C to 55.8 °C, suggesting that c-di-GMP stabilizes the protein, while c-di-AMP had no effect (Fig. 1b). In addition, we used differential radial capillary action of ligand assays (DRaCALA), which allows visualization of protein-bound radiolabeled ligand as a central spot after the application of the protein-ligand mix onto nitrocellulose[41]. Using DRaCALA we confirmed

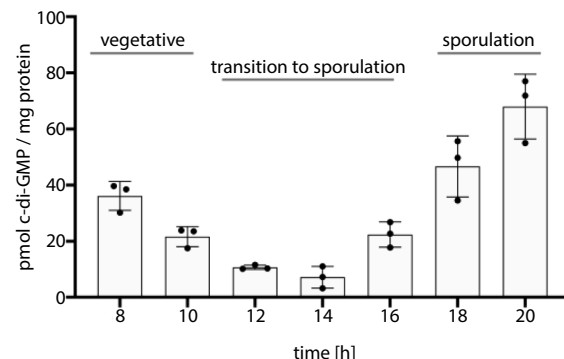

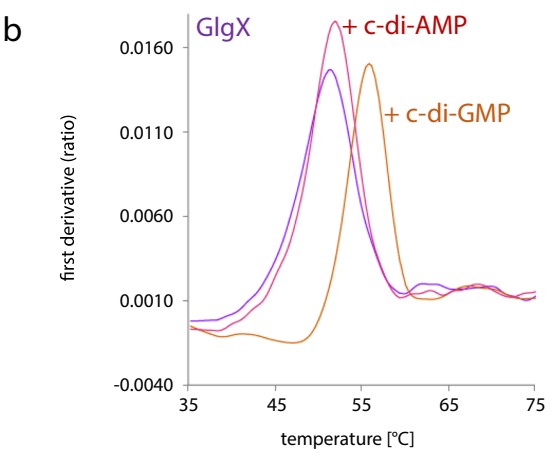

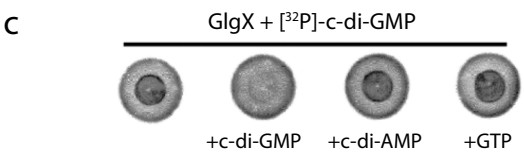

c

GlgX + [³²P]-c-di-GMP

+c-di-GMP  +c-di-AMP  +GTP

**Fig. 1 | GlgX binds c-di-GMP. a** Quantification of c-di-GMP in *S. venezuelae* cell extracts using LC-MS/MS. Cells were harvested during late vegetative growth (8–10 h), transition to sporulation (12–16 h), and sporulation (18–20 h). Data are presented as the mean of biological replicates ±SD (*n* = 3). Source data are provided in the Source data file. **b** nanoDSF thermal shift first-derivative curves of 10 μM GlgX without any ligand (purple), with 1 mM c-di-AMP (pink), or 1 mM c-di-GMP (orange). c-di-GMP increases the melting point of GlgX from 51.5 to 55.8 °C. Source data are provided in the Source data file. **c** [³²P]-c-di-GMP binds to GlgX in DRaCALA assay indicated by dark spots in the center of the nitrocellulose membrane. For competition, excess (100 μM) of unlabeled c-di-GMP, c-di-AMP, or GTP was added to the reaction containing [³²P]-c-di-GMP and GlgX.

that GlgX from *S. venezuelae* binds ³²P-labeled c-di-GMP. Moreover, excess unlabeled c-di-GMP but not c-di-AMP or GTP competed with ³²P-c-di-GMP for binding to GlgX, demonstrating that binding of c-di-GMP is specific (Fig. 1c). We also tested the *S. venezuelae* GlgX paralog, His₆-Vnz25270, and His₆-GlgX$_{EC}$ using nano DSF and DRaCALA, respectively, but did not detect any interactions with c-di-GMP by these proteins (Supplementary Fig. 4a–c).

Arginine and aspartic acid arranged in an RxxD motif are often found as c-di-GMP binding signatures in effector proteins[28,34]. Examination of the GlgX sequence revealed two RxxD motifs, RETD$_{49-52}$ and RLVD$_{212-215}$. Notably, these motifs are missing in Vnz25270 and GlgX$_{EC}$,

which do not bind c-di-GMP (Supplementary Fig. 4). To determine if either of these motifs are involved in c-di-GMP binding, we swapped the arginine and aspartic acid residues in the two motifs and purified His₆-GlgX$_{DETR(49-52)}$ and His₆-GlgX$_{DLVR(212-215)}$. NanoDSF experiments revealed that the presence of c-di-GMP had no effect on His₆-GlgX$_{DETR(49-52)}$ and DRaCALA assays support that the His₆-GlgX$_{DETR(49-52)}$ mutant does not bind c-di-GMP (Supplementary Fig. 5a). By contrast, c-di-GMP still caused a shift in the melting/unfolding temperature of the His₆-GlgX$_{DLVR(212-215)}$ protein variant (Supplementary Fig. 5b). Collectively, these data indicate that GlgX from *S. venezuelae* is a c-di-GMP effector and that the RxxD motif at position 49-52 is involved in ligand binding.

## c-di-GMP induces GlgX-mediated glycogen degradation

Glycogen debranching enzymes hydrolyze α-(1→6)-linked glucan branches, producing malto-oligosaccharides with reducing ends[7]. The bicinchoninic acid (BCA) assay allows accurate photometric determination of reducing ends from oligosaccharides and is based on the reduction of Cu²⁺ to Cu⁺. A purple copper bicinchoninate complex with an absorbance maximum at 562 nm forms from Cu⁺ and bicinchoninate[42]. Using the BCA assay, we found that increasing concentrations of c-di-GMP (3.125–100 μM) induce GlgX-mediated generation of reducing ends in glycogen from rabbit, with saturating enzymatic activity at 50 μM c-di-GMP (Fig. 2a). In addition, we tested the effect of c-di-GMP on GlgX activity by using polysaccharides with different degrees of α-(1→6)-linked glucans as a substrate (Fig. 2b). While glycogen contains α-(1→6)-linked glucan branches every 8–10 residue, amylose is essentially an unbranched chain of α-1,4-linked glucose subunits. Amylopectin is composed of α-1,4-linked glucose units with α-(1→6)-linkages at intervals of ~20 units[43]. Pullulan is a polysaccharide polymer consisting of maltotriose units in which the three glucose units are connected by α-1,4-glycosidic bonds. Consecutive maltotriose units are linked by a α-1,6-glycosidic bond. By combining 0.05% of the respective carbohydrate with 1 μM His₆-GlgX in the absence or presence (50 μM) of c-di-GMP we found that c-di-GMP stimulates GlgX to hydrolyze glycogen, pullulan and amylopectin, while no stimulation of GlgX activity could be detected when amylose was used as a substrate (Fig. 2b).

In the *S. solfataricus* GDE, called TreX, the catalytic triad was mapped to two aspartic acids and one glutamic acid which correspond to Asp342, Glu378, and Asp450 in *S. venezuelae* GlgX (highlighted in yellow in Supplementary Fig. 2)[38]. We mutagenized selected active site residues and tested His₆-GlgX$_{D342A}$, His₆-GlgX$_{E378A}$ and His₆-GlgX$_{D342A+E378A}$ for activity in BCA assays. In line with our expectations, we found that enzyme activity of mutagenized GlgX could not be stimulated by the addition of c-di-GMP, when compared to the wild type (WT) GlgX. Moreover, in agreement with our c-di-GMP binding studies (Supplementary Figs. 4 and 5), our data demonstrate that the hydrolytic activity of neither His₆-Vnz25270 nor of the c-di-GMP non-binding variant His₆-GlgX$_{DETR49-52}$ was affected by c-di-GMP and remained at basal levels (Fig. 2c).

Next, we investigated if GlgX affects glycogen accumulation in vivo in *S. venezuelae*. To address this question, we generated a *glgX* null mutant by the redirect PCR targeting approach[44] and a strain overexpressing *glgX* from the constitutive *ermE*\* promoter from the integrative pIJ10257 vector[45]. Transmission electron microscopy of five-day old colonies stained for α-glucans revealed that spores of Δ*glgX* accumulated more glycogen than WT spores. Only few glycogen-containing spores were found in the strain overexpressing *glgX* (Fig. 3a). In line with this observation, we determined 817.4 ± 141.5 and 193.7 ± 17.8 μg glycogen/mg protein in the Δ*glgX* mutant and WT, respectively. Complementation of the Δ*glgX* mutant with the FLAG-tagged allele restored glycogen content to WT levels (173.1 ± 13.4 μg/mg protein). Overexpression of *glgX* caused a drop of glycogen to 19.8 ± 7.8 μg/mg protein (Fig. 3b).

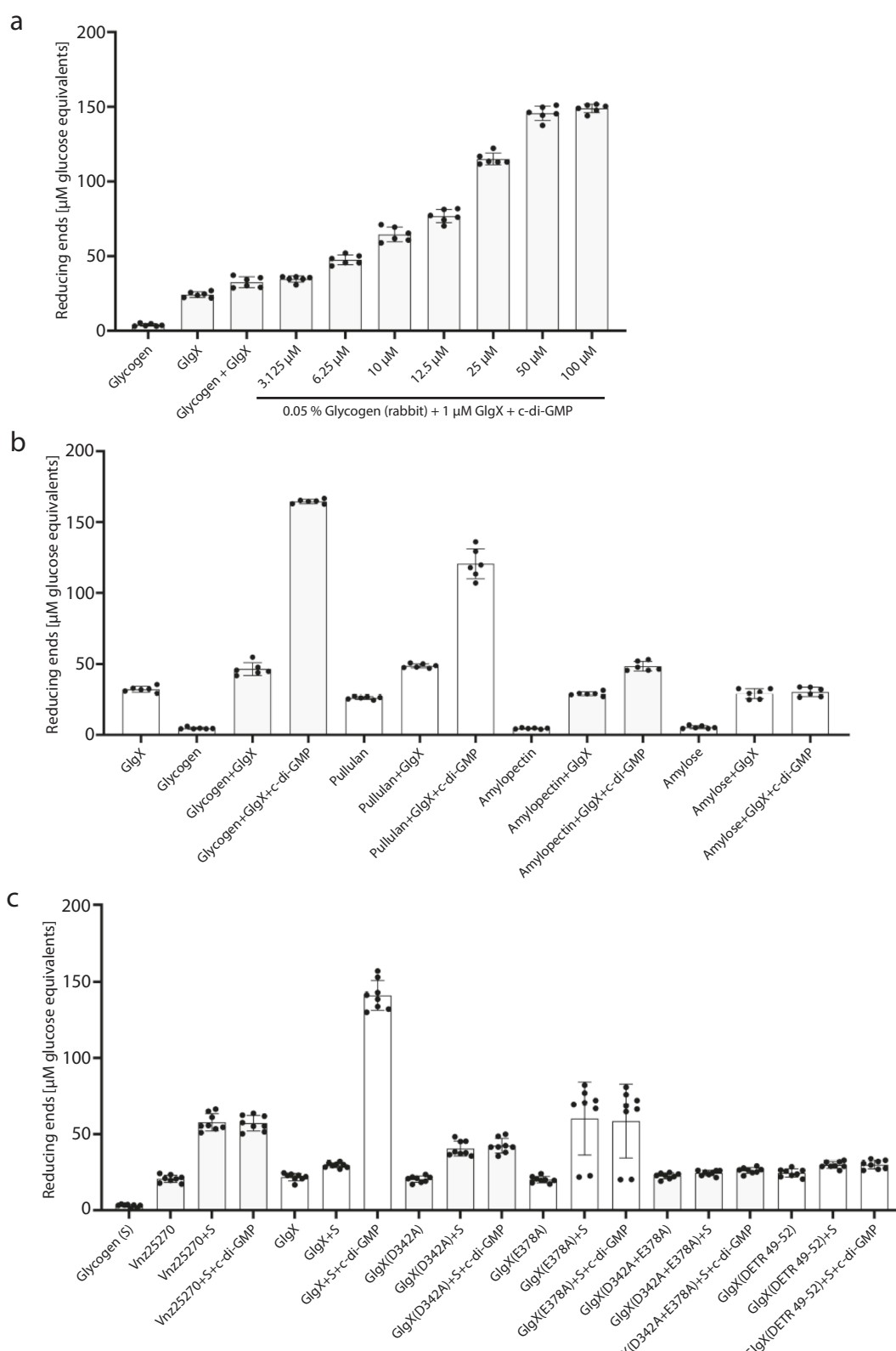

*Streptomyces* spore pigments are aromatic polyketides that are produced by enzymes encoded in the highly conserved *whiE* gene cluster. The expression of *whiE* genes is developmentally regulated and the synthesis of the spore pigment represents one of the last events in spore maturation[30,46]. Although we did not detect an effect of *glgX* deletion on formation of mature spores as judged by the formation of the green spore pigment, overexpression of *glgX* caused a white phenotype, indicating a defect in spore maturation (Fig. 3c). Moreover, our quantification of spores revealed that *S. venezuelae* overexpressing *glgX* produces a reduced number of viable spores compared to the WT or the Δ*glgX* mutant, respectively (Supplementary Fig. 6). Thus, our data demonstrate that GlgX functions to break down glycogen in a c-di-GMP stimulated manner and that Asp342, Glu378, and Asp450 are important for its enzymatic activity.

**Fig. 2 | c-di-GMP stimulates glycogen breakdown activity of GlgX in vitro.**
**a** Bicinchoninic acid (BCA) in vitro enzyme assay. The purple copper bicinchoninate complex indicating GlgX-mediated generation of reducing ends in glycogen was quantified by photometric measurements at 562 nm. $A_{562}$ values were converted into reducing ends [μM glucose equivalents] based on ᴅ-glucose as a standard. Reactions contained 0.05% (w/v) glycogen from rabbit, 1 μM GlgX and increasing concentrations of c-di-GMP (0–100 μM). Source data are provided as a Source data file. **b** BCA assay containing 0.05% of either glycogen from rabbit, pullulan,

amylopectin from maize or amylose from potato as substrates. Where appropriate, 1 μM GlgX or 50 μM c-di-GMP were added to the reaction samples. Source data are provided as a Source data file. **c** BCA assay as in **a** with or without c-di-GMP (50 μM), 0.05% glycogen from rabbit (S) and 1 μM Vnz25270, GlgX, the active site mutants GlgX$_{D342A}$, GlgX$_{E378A}$, GlgX$_{D342A+E378A}$, or the c-di-GMP-binding mutant variant GlgX$_{DETR49-52}$ as indicated. The experiment was performed three times, each time with two technical replicates. The data points represent the mean ± SD of six data points. Source data are provided as a Source data file.

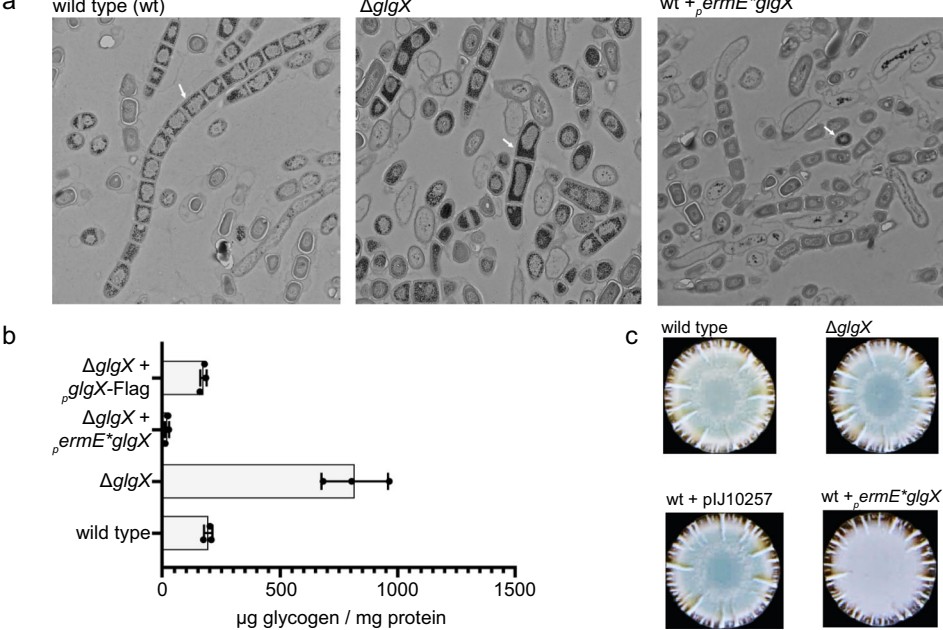

**Fig. 3 | Δ*glgX* mutant accumulates increased levels of glycogen.** **a** Transmission electron micrographs of colonies stained for α-glucans of *S. venezuelae* WT, Δ*glgX* and a strain overexpressing *glgX* from the constitutive *ermE*\* promoter. Strains were grown for 5 days on MYM agar at 30 °C. **b** The glycogen content of 5 days-old colonies of *S. venezuelae* WT, Δ*glgX*, Δ*glgX* complemented with FLAG-tagged *glgX* expressed from the native promoter and Δ*glgX* overexpressing *glgX* from the *ermE*\*

promoter was quantified using the glycogen assay kit (MAK016) from Sigma-Aldrich. Data are presented as a mean of three biological repeats +/- SD. **c** Macro-colonies of *S. venezuelae* WT, Δ*glgX*, and WT either containing the empty pIJ10257 vector or expressing *glgX* from the strong *ermE*\* promoter. Strains were grown on MYM-agar containing 0.03% maltose for 48 h.

## Crystal structure of *S. venezuelae* GlgX

To gain insight into *S. venezuelae* GlgX function and how it binds and is activated by c-di-GMP we performed structural studies. We first obtained a crystal structure of the apo GlgX enzyme to 3.5 Å resolution (Methods section). The structure was refined to $R_{work}/R_{free}$ values of 21.1%/27.6% (Supplementary Table 1 and Supplementary Fig. 7) and consists of three main domains, an N-terminal β-sandwich region (residues 1–145) comprised of eight β-strands and several short helices, a Central domain (residues 146–575) that harbors a α/β barrel fold similar to that found in many α-amylase enzyme structures and which consists of eight β-strands surrounded by eight α-helices. Finally, the C-terminal domain (residues 576–706) contains three β-sheets assembled from eight β-strands (Fig. 4a). Only a few GDE structures have been solved and structural homology searches revealed the *S. venezuelae* GlgX structure shows the strongest similarity to the TreX GDE structure from *S. solfataricus*; the structures can be overlaid with a root mean square deviation (rmsd) of 1.4 Å for 641 Cα atoms[38] (Fig. 4b).

## Structure of the *S. venezuelae* GlgX-c-di-GMP complex

To elucidate the molecular mechanism of c-di-GMP binding by GlgX we crystallized the *S. venezuelae* GlgX-c-di-GMP complex and solved the structure to 3.34 Å resolution. The structure was solved by using the *S. venezuelae* apo GlgX as a search model (Methods section) and the structure was refined to final $R_{work}/R_{free}$ values of 20.5%/27.5%

(Supplementary Table 1, Supplementary Fig. 7 and Fig. 5a, b). Strikingly, in the structure, a c-di-GMP monomer is bound at each end of an antiparallel GlgX dimer (Fig. 5a, b). Note, a similar dimer was observed in the crystal packing of the apo GlgX.

The c-di-GMP molecules, bound at each end of the GlgX molecule, make contacts to GlgX residues 31–51 from the N-domain of one GlgX subunit and residues 430–437 and 628–638 from the Central and C-domain of the other GlgX subunit (Fig. 5a, b). Residues in the GlgX N-domain provide most of the base specifying contacts with the c-di-GMP. In particular, the N1 and N2 atoms of one guanine base are contacted by the side chain of Glu47 while the side chain of Arg56 makes a hydrogen bond with the O6 atom of the guanine (Fig. 5b). Cation-π interactions to this guanine base are provided by Arg31 and Arg49. Specifying contacts to the other guanine base are primarily from GlgX main chain atoms. The carbonyl oxygen of Thr51 hydrogen bonds with the guanine N1 atom and the Thr51 amide nitrogen contacts the guanine O6 atom (Fig. 5b). There are also two hydrogen bonds from residues in the Central domain of the other subunit to this guanine base; the main chain amide nitrogen of Gly436 hydrogen bonds to the guanine N2, while the side chain of Arg438 hydrogen bonds with the O6 atom of the guanine and also stacks with the guanine base along with the side chain of Arg579 (Fig. 5b). The two c-di-GMP phosphates are contacted from the side chains of Ser629 and Gln433. Notably, the observed c-di-GMP contacts in our structure are

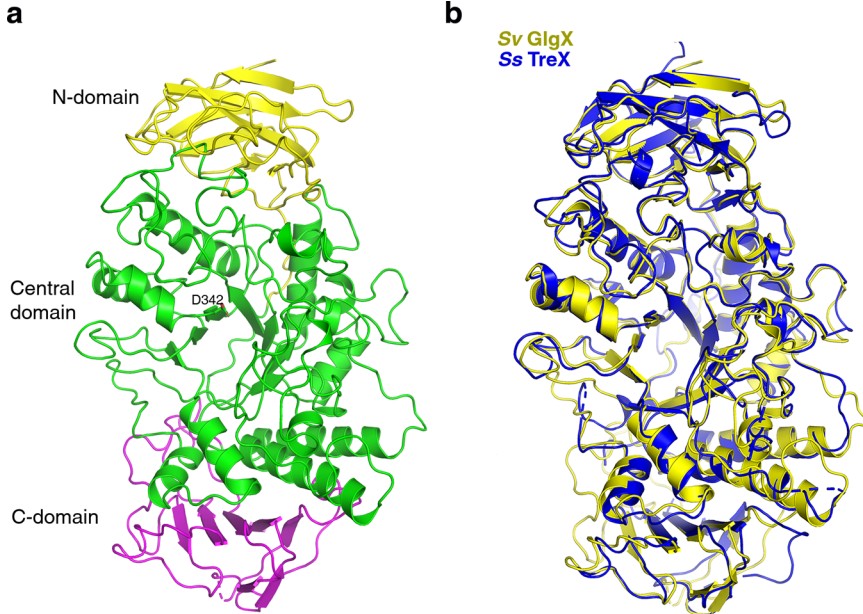

**a**

N-domain

Central domain

D342

C-domain

**b**

*Sv* GlgX
*Ss* TreX

**Fig. 4 | Crystal structure of apo *S. venezuelae* GlgX. a** Ribbon diagram of the *S. venezuelae* GlgX. The domains are labeled and shown in different colors. The N-domain is colored yellow, the Central domain is green and the C-domain is magenta. Also shown as sticks and labeled is the position of the active site residue,

Asp342. This figure and other ribbon diagrams were generated using PyMOL (the PyMOL Molecular Graphics System, Version 2.0 Schrödinger, LLC). **b** Overlay of the *S. venezuelae* GlgX (yellow) and TreX (blue) structures showing they have similar overall domain arrangements but large differences in loop regions.

consistent with our biochemical data showing that the RETD$_{49-52}$ motif participates in c-di-GMP binding. The collective interactions from GlgX completely specify each of the guanine bases and would select against binding c-di-AMP.

The GlgX-c-di-GMP structure revealed that one c-di-GMP ribose hydroxyl group is exposed to solvent indicating that GlgX should be capable of binding 2'-O-(6-[Fluoresceinyl]aminohexylcarbamoyl)-cyclic diguanosine monophosphate (2'-Fluo-AHC-c-di-GMP) (herein called F-c-di-GMP), which has a fluorescein tag attached to one ribose hydroxyl group. Thus, we used this F-tagged c-di-GMP in fluorescence polarization (FP) studies to quantify the interaction between GlgX and c-di-GMP. These experiments revealed a $K_d$ of $8.2 \pm 1.0\,\mu$M for the interaction (Fig. 5c). In addition, consistent with our structure, GlgX showed essentially no binding to the similarly labeled F-c-di-AMP probe. To ensure that the presence of the fluorescent tag was not impacting the GlgX interaction with c-di-GMP we used microscale thermophoresis (MST) to quantify c-di-GMP binding to GlgX. This experiment produced a $K_d$ of 8.3 $\mu$M (Supplementary Fig. 8), which is in line with our FP data. As shown above, our analyses revealed that the RxxD swap mutant abrogated saturable c-di-GMP binding consistent with our structure. But to further probe the structural model we created GlgX$_{E47A-R49A}$ and GlgX$_{R438A-R579A}$ mutants and performed FP studies with these proteins. Both mutant proteins showed weak, nonsaturable binding to c-di-GMP (Fig. 5c), supporting the structural model.

Our data indicate that c-di-GMP is required for GlgX function. However, the c-di-GMP binding site is ~30 Å from the putative GlgX active site (Fig. 6), raising the question of how this interaction could impact GlgX enzyme activity. Insight into this question was provided by an overlay of the c-di-GMP bound *S. venezuelae* GlgX onto the apo structure. This overlay revealed that significant conformational changes occur upon c-di-GMP binding in several regions of the structure that are transmitted to the active site region (Fig. 6). These conformational changes result from interaction of the c-di-GMP with the C-domain of one GlgX subunit as c-di-GMP contacts with the N-domain elicits little structural perturbations (Fig. 6). Specifically, c-di-GMP binding leads to a restructuring of two loops formed by residues

432–442 and 579–593 within the C-domain region bound by the second messenger. This results in concomitant conformational changes in residues 377–396 and residues 341–355. The alterations in the latter includes the unfolding of two helical regions. Importantly, residues 377–396 and 341–355 contain the key catalytic residues, Glu378 and Asp342. Hence, the conformational changes lead to a restructuring of the active site pocket (Fig. 6 and Supplementary Fig. 9). Thus, these analyses indicate that c-di-GMP binding acts as an allosteric regulator of GlgX to stabilize its active state. But to further test this hypothesis, we next crystallized *S. venezuelae* GlgX with c-di-GMP and the substrate analog, acarbose.

### Structures of *S. venezuelae* GlgX-c-di-GMP-acarbose complexes

To visualize the effects of c-di-GMP binding on GlgX substrate binding and catalysis we crystallized the GlgX-c-di-GMP complex in the presence of substrate analog, acarbose (Methods section). Acarbose is a pseudo-tetrasaccharide that is a well-known amylase inhibitor. Two crystal structures of *S. venezuelae* GlgX-c-di-GMP-acarbose were obtained (Supplementary Table 1) one at pH 4.6 and the other, pH 8.5. The GlgX-c-di-GMP-acarbose structures solved at pH 8.5 and pH 4.6 were refined to $R_{work}/R_{free}$ values of 23.6%/30.3% to 3.6 Å and $R_{work}/R_{free}$ values of 17.2%/23.4% to 2.4 Å resolution, respectively. Both structures revealed clear density for c-di-GMP molecules bound in the same locations. Interestingly, however while the pH 4.6 structure revealed density for all 4 rings of the acarbose docked in the active site, the pH 8.5 structure captured an intermediate in catalysis whereby the reducing end of the molecule had been cleaved resulting in the loss of one sugar and density at −1 of the reducing end of the molecule (using nomenclature from Davies et al.)[47]. GlgX activity is likely inhibited at pH 4.5, explaining why the acarbose is unreacted in that complex, while the GlgX-c-di-GMP-acarbose structure obtained at pH 8.5 allowed us to capture a covalent intermediate[38].

Comparison of the GlgX active site conformation in the GlgX-c-di-GMP structure to the acarbose bound structures reveals that they are essentially identical. These data therefore support that c-di-GMP acts as a positive allosteric effector by favoring the enzymatically active conformation of the enzyme (Fig. 7a–c). Indeed, the position of the

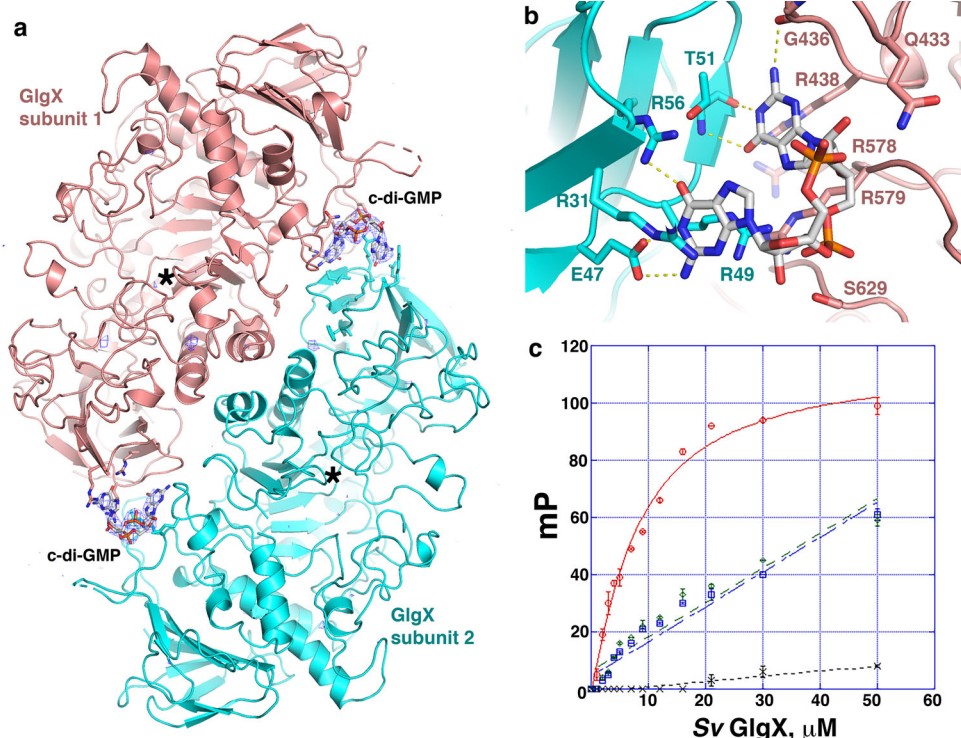

**Fig. 5 | Structure of the _S. venezuelae_ GlgX-c-di-GMP complex. a** Ribbon diagram of the _S. venezuelae_ GlgX-c-di-GMP complex with one GlgX subunit colored cyan and the other salmon. Shown as sticks are the c-di-GMP molecules bound at each end of the antiparallel dimer. Also displayed over the structure is the mFo-DFc omit map, shown as a blue mesh and contoured at 4.2 σ in which the c-di-GMP molecules had been removed prior to multiple rounds of refinement. There was clear density for each c-di-GMP molecule. **b** Close up of the GlgX-c-di-GMP binding interaction. Key residues mediating contacts to the c-di-GMP are shown as sticks and labeled. Hydrogen bonds from GlgX residues that specify binding to the guanine are indicated by yellow dashed lines. **c** Fluorescence polarization (FP) binding isotherm showing interaction of GlgX, GlgX$_{E47A\cdot R49A}$ and GlgX$_{R438A\cdot R579A}$ for F-c-di-GMP in red, green and blue respectively. Also shown is the binding isotherm for GlgX binding to F-c-di-AMP in black. WT GlgX-bound F-c-di-GMP with a $K_d$ of $8.2 \pm 1.0$ μM while the mutants showed nonsaturable binding and WT GlgX also did not bind F-c-di-AMP. The _y_-axis and _x_-axis are millipolarization (mP) units and GlgX concentration (μM), respectively. Shown are representative binding curves for each. The experiments were performed in technical triplicates and the error between measurements reported. Error bars represent SD of the measurements. Source data are provided in the Source data file.

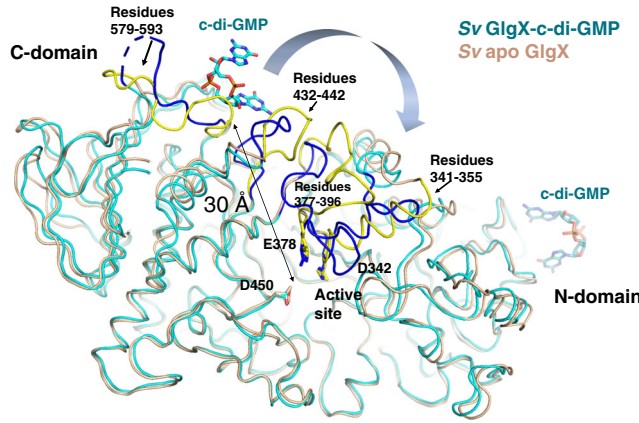

**Fig. 6 | c-di-GMP induces long-range conformational changes in _S. venezuelae_ GlgX impacting the active site.** Shown is an overlay of similar Cα atoms of apo GlgX (tan) onto the GlgX-c-di-GMP structure (cyan). The c-di-GMP bound in the GlgX-c-di-GMP structure is also colored cyan. Structural changes upon c-di-GMP binding occur in the C-terminal c-di-GMP binding region with residues 579–593 and 432–442 shifting in position. These structural changes lead to reorganization of residues 377–396 and residues 341–355, both of which contain key catalytic residues. The regions that undergo conformational changes are colored blue for the c-di-GMP bound state and yellow for the apo state. This is how c-di-GMP, which binds 30 Å from the active site, can induce changes that result in reorganization of the active site.

substrate binding residues in the GlgX-c-di-GMP structure superimpose onto the structure with bound acarbose, including the arrangement of Asp342, Glu378, and Asp450 (Fig. 7b). Key contacts to the −1 substrate sugar are provided by GlgX residues Trp222, Tyr224, His271, Asp342, Glu378, W380, His449 as well as Asp342. The remaining sugar moieties are contacted by Arg212, Glu470, Asn534, and Tyr536 (Fig. 7b). The c-di-GMP contacts are essential for catalysis as they trigger the structural changes leading to the active conformational state through the C-domain (Fig. 6).

To date _S. venezuelae_ GlgX is the only GDE identified that binds c-di-GMP. The structure reveals key residues for c-di-GMP binding that can be used to interrogate other GDEs to assess whether they may bind c-di-GMP. Sequence alignments of GlgX with the TreX protein shows that the latter enzyme lacks key c-di-GMP binding residues, including Arg52, Arg56, Arg438, and Arg579 indicating that TreX would not bind c-di-GMP (Supplementary Fig. 2). But while c-di-GMP binding residues are also not conserved in Vnz25270 and GlgX$_{EC}$, consistent with our data showing that they do not bind c-di-GMP, sequence alignments of _S. venezuelae_ GlgX with GDE enzymes from other streptomycetes species reveals that key c-di-GMP binding residues are conserved amongst these proteins (Supplementary Table 2) indicating a conserved mode of c-di-GMP regulation of these enzymes among _Streptomyces_.

## Discussion

c-di-GMP signals through two master transcriptional regulators, BldD and the σ factor, WhiG, respectively, to control the two key transitions of the _Streptomyces_ life cycle: the formation of reproductive aerial

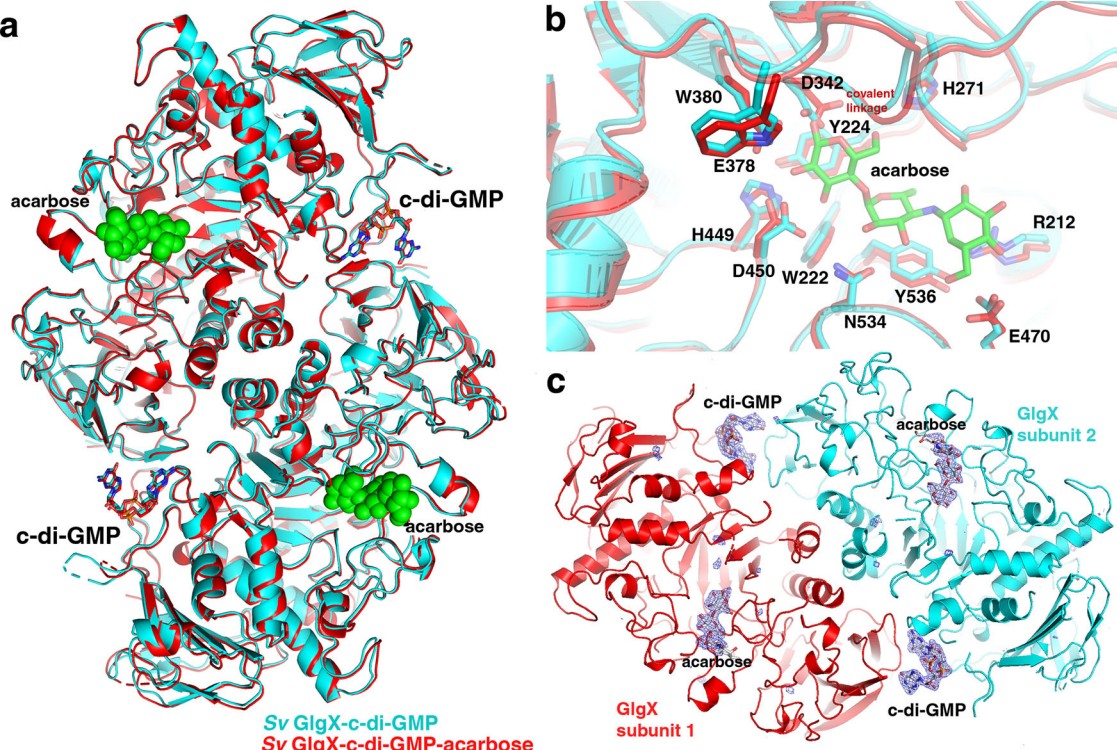

**Fig. 7 | Structures of *S. venezuelae* GlgX-c-di-GMP-acarbose complexes.**
**a** Structure of the *S. venezuelae* GlgX-c-di-GMP-acarbose complex (pH 8.5) (red) superimposed onto the *S. venezuelae* GlgX-c-di-GMP complex (cyan). The acarbose molecules are bound in the active site and shown as green spheres. The c-di-GMP molecules are bound at the ends, in the identical location in both structures.
**b** Close up of the bound acarbose in the *S. venezuelae* GlgX-c-di-GMP-acarbose complex (red). Shown are the side chains of the residues that contact the acarbose

in both the c-di-GMP-acarbose (red) and c-di-GMP bound (cyan) structures.
**c** Structure of the *S. venezuelae* GlgX-c-di-GMP-acarbose complex solved at pH 4.5. In this structure, the acarbose has not reacted with GlgX (as density is evident for all sugar moieties). One GlgX subunit is cyan and the other red, the c-di-GMP and acarbose molecules are shown as sticks. The omit $mF_o-DF_c$ map in which the ligands had been removed is also shown as a blue mesh and contoured at 4σ.

hyphae and their differentiation into spore chains. In both these cases, the second messenger acts as a brake to developmental transition. BldD controls the first developmental switch, whereby it sits at the top of the developmental regulatory network, serving to repress a large set of genes, including many genes encoding downstream transcriptional regulators. BldD binds a unique tetrameric cage of c-di-GMP that acts as a BldD dimerizer, enabling its binding to palindromic DNA targets[33–35]. The release of c-di-GMP from BldD thus derepresses genes allowing the transition from the vegetative mycelium stage. C-di-GMP also controls the transition of *Streptomyces* from aerial hyphae to sporulation. In this case, c-di-GMP mediates complex formation between the key sporulation σ factor, WhiG, and its anti-σ, RsiG. The dissociation of c-di-GMP from the RsiG-WhiG complex releases WhiG allowing it to activate key sporulation genes. For proper *Streptomyces* development, c-di-GMP must dissociate from BldD prior to WhiG being released from RsiG. Such an order of events is consistent with data showing that BldD binds c-di-GMP with lower affinity ($K_d$ of $2.5 \pm 0.6\,\mu M$) than the WhiG-RsiG complex ($K_d$ for [RsiG + WhiG] is $0.39 \pm 0.05\,\mu M$)[34,35].

This ordered model would necessitate that there is a drop in c-di-GMP levels in *Streptomyces* during development, however, this has never been demonstrated. Here we show that indeed, c-di-GMP levels are high during the initial stages of *Streptomyces* development and then diminish during the transition from the vegetative mycelium stage to sporulation (Fig. 1a). Interestingly, we found that during sporulation c-di-GMP levels again rise significantly, which may be needed to fortify the newly formed spores for the next round of germination and vegetative growth. Driven by the hypothesis that heretofore uncharacterized c-di-GMP effectors may be sensing this signal

at this later stage in development, we identified the glycogen debranching enzyme, GlgX, as a c-di-GMP binding protein. Interaction between GlgX and c-di-GMP is likely favored only during the late developmental stage for two reasons. First, the protein is particularly abundant during sporulation (Supplementary Fig. 3). Second, GlgX has a relatively low affinity for c-di-GMP (~8 μM; see Fig. 5c and Supplementary Fig. 8) and therefore needs higher levels of c-di-GMP that are detectable in the late growth stages (Fig. 1a) for binding.

We found that not only does GlgX bind c-di-GMP, but this interaction stimulates the catalytic activity of the enzyme to hydrolyze glycogen (Fig. 2). To elucidate the molecular basis for this unusual mode of c-di-GMP mediated enzyme activation, we solved a series of *S. venezuelae* GlgX crystal structures, including the apo GlgX, GlgX-c-di-GMP complex and the GlgX-c-di-GMP complex with substrate acarbose (Figs. 4, 5, and 7). The *S. venezuelae* GlgX-c-di-GMP structure revealed that binding the second messenger stabilizes the active enzyme conformation, thus explaining how it acts as an allosteric regulator (Fig. 6).

In the structure, c-di-GMP binds in an asymmetric, surface exposed pocket that is formed between two GlgX subunits; the pocket is forged by residues from the N-domain, including $RxxD_{49-52}$ from one GlgX subunit and residues from the central and C-domain from the other GlgX subunit. The structures of several c-di-GMP binding proteins complexed to c-di-GMP have recently been solved and reveal a diverse range of binding modes and motifs[28,48,49]. The RxxD motif was the first c-di-GMP binding motif discovered and has since emerged as one of the most common motifs for binding the second messenger[48,49]. For example, the RxxD motif is found in several autoinhibitory (I) sites in active and inactive GGDEF DGCs and usually precedes the GGDEF motif. Structures of RxxD motif containing proteins bound to c-di-

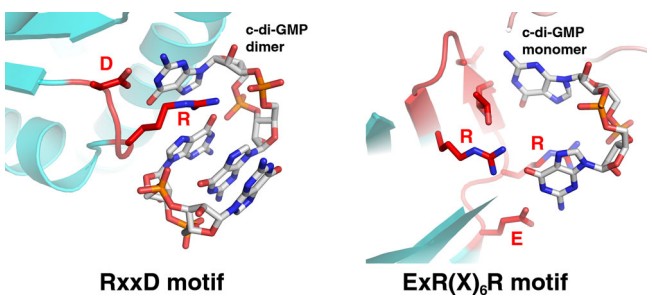

**Fig. 8 | Comparison of c-di-GMP bound to a canonical RxxD motif (left) in the PelD-c-di-GMP complex (4DN0: https://www.rcsb.org/structure/4dn0) to the RxxD containing region in the *S. venezuelae* GlgX-c-di-GMP complex (right).** The arginine and aspartic acid of the PelD RxxD motif specifies two bases of a bound c-di-GMP intercalated dimer. By contrast, the guanine bases of the c-di-GMP bound to GlgX are specified by residues outside the arginine and aspartic acid of the GlgX RxxD region, specifically the first glutamic acid and last arginine in the GlgX ExR(X)$_6$R motif as well as main chain atoms.

GMP have revealed that in these proteins the RxxD motif adopts a β-turn loop conformation with the side chains of the conserved arginine and aspartate residues making specific hydrogen bonds with c-di-GMP guanine bases. RxxD motifs in these proteins also typically interact with c-di-GMP intercalated dimers.

Interestingly, although GlgX contains a RxxD sequence, it utilizes a very different mode of second messenger recognition from previously studied proteins that bind c-di-GMP with RxxD motifs (Fig. 8). First, the RxxD sequence in GlgX binds a monomeric form of c-di-GMP. Second, this motif in GlgX is located in a loop between β strands and it adopts a different conformation from the β-turn loop conformation. Further, residues in the GlgX RxxD sequence makes different contacts to c-di-GMP compared to these proteins. In particular, while Arg49 from the GlgX RxxD motif contacts c-di-GMP, it stacks with a guanine base but does not make specifying hydrogen bonds. In addition, the aspartic acid in the GlgX RxxD motif, Asp52, not only does not contact a guanine base, but it makes no contacts with the c-di-GMP. Instead, Asp52 interacts with the side chain of Arg438, helping to orient it properly for hydrogen bonding and stacking interactions with a c-di-GMP guanine base. Thus, our GlgX structures uncover a c-di-GMP binding motif that would be better defined as an ExR(X)$_6$R motif, which highlights the residues involved in base contacts, Glu47 and Arg56 (Fig. 8). This finding also underscores the challenge in predicting the details of c-di-GMP interactions based on sequence alone.

The strict conservation of key c-di-GMP binding residues among *Streptomyces* (Supplementary Table 2 and Supplementary Fig. 10) suggests this interaction and resultant enzymatic activation plays a key role in the development of these bacteria, when they may need a burst in fast metabolizing energy sources.

## Methods

### Bacterial strains and growth conditions

All strains used in this study are listed in Supplementary Table 3. *E. coli* strains were grown in LB medium under aerobic conditions at 37 °C. If relevant, LB was supplemented with 100 μg/ml ampicillin (Amp), 50 μg/ml kanamycin (Kan), 50 μg/ml apramycin (Apr), and/or 15 μg/ml chloramphenicol (Cam). 16 μg/ml and 22 μg/ml hygromycin B (Hyg) were added to Nutrient Agar (NA; Roth) and LBon (LB without salt), respectively, when needed. For nucleotide extraction, biological triplicates of WT *S. venezuelae* (Supplementary Table 3) were grown aerobically at 30 °C and 170 rpm in liquid Maltose-Yeast extract-Malt extract (MYM) medium[50] containing 50% (v/v) tap water and 0.2% (v/v) trace element solution[51] for 20 h. For each replicate, 100 ml MYM were inoculated with spores at a final concentration of 10$^6$ CFU/ml. After initial outgrowth for 8 h, samples for nucleotide extraction (5 ml),

protein quantification (1 ml), microscopic analysis (100 μl) and measurements of optical density (1 ml) were taken every two h. Fragmentation was detectable after 12–14 h of growth, spores became visible after 16–18 h of incubation. Cells were imaged using the Leica DM2000 LED microscope at 100× magnification (Supplementary Fig. 1a). Optical density was measured at 578 nm (Supplementary Fig. 1b).

To analyze *S. venezuelae* development, 12 μl spores (2.4 × 10$^6$ CFU) in 20% (v/v) glycerol were spotted on MYM agar containing 0.03% (w/v) maltose and macrocolonies were photographed using a Canon EOS 1300D (W) camera after 48 h of growth at 30 °C. For quantification of viable spores, the spore stock of the respective *S. venezuelae* strain was adjusted to OD$_{600}$ of 2.5 and 10 μl of a 1:1000 dilution were spotted on MYM agar containing 0.03% (w/v) maltose and 50 μg/ml hygromycin. Macrocolonies were grown for 48 h at 30 °C. For each of the 3 biological replicates, 4 macrocolonies were scraped from the plate and transferred into 2 ml 20% (v/v) glycerol. After vortexing for 1 min, the samples were incubated for 5 min at room temperature (rt). 1.5 ml of the suspension was centrifuged for 2 min at 1000 × *g* and 1 ml of the supernatant containing the spores was filtered using a 5 μm filter. Appropriate dilutions were plated on LB plates and colonies were counted after 2 days incubation at 30 °C.

### Construction of plasmids

The oligonucleotides used for plasmid constructions are listed in Supplementary Table 3. pIJ10257_glgX was constructed to generate *glgX* overexpression strains in *S. venezuelae*. *glgX* was amplified from *S. venezuelae* genomic DNA using the primer pair MW87/187. The PCR product was cloned into NdeI/HindIII restriction sites of pIJ10257, which places *glgX* under the control of the constitutive *ermE*\* promoter. p3xFLAG_glgX was constructed to analyze protein expression along the developmental life cycle in *S. venezuelae* and for complementation of the *glgX* mutant, respectively. The *glgX* gene including ~300 bp upstream of the open reading frame covering the promoter region was amplified from *S. venezuelae* genomic DNA using primer pair MW171/172. The PCR product was cloned into HindIII/XhoI sites of p3xFLAG. For overexpression and purification of N-terminally His$_6$-tagged GlgX protein variants, the respective genes were cloned into the pET15b vector (Novagen). For the construction of pET15b_glgX and pET15b_vnz25270, the corresponding genes were amplified from *S. venezuelae* genomic DNA using primer pair MW87/88 and MW185/186. *E. coli* genomic DNA served as a template for the amplification of *glgX$_{E.coli}$* using the primers MW190/191. pET15b_glgX$_{D342A}$ and pET15b_glgX$_{E378A}$ were constructed by inverse PCR using pET15b_glgX as template plasmid and primer pair MW197/198 and MW199/200, respectively. pET15b_glgX$_{D342A+E378A}$ was generated by inverse PCR using pET15b_glgX$_{D342A}$ as template plasmid and primer pair MW199/200. For the generation of pET15b_glgX$_{DETR(49-52)}$ and pET15b_glgX$_{DLVR(212-215)}$, pET15b_glgX was used as a template combined with the primer pairs MW213/205 and MW206/214 or MW213/209 and MW210/214, respectively. The pET15b backbone was amplified from pET15b_glgX using primer pair MW215/216. Respective DNA fragments were assembled using Gibson assembly. Sequences of all inserts were verified by sequencing at LGC Genomics Berlin.

### Generation of *S. venezuelae* Δ*glgX* mutant

Gene deletion in *S. venezuelae* was performed using a modified Redirect PCR targeting protocol described in[44]. In these experiments, *E. coli* BW25113 containing the λ RED plasmid pIJ790 was transformed with the PI1_F8 cosmid and grown in LB under aeration at 30 °C. Next, BW25113/pIJ790 carrying PI1_F8 was transformed with a PCR fragment containing the *apr-oriT* cassette flanked by homologous regions to the *glgX* locus. Respective homologous sequences were fused to the *apr-oriT* cassette by PCR amplification using primer pair MW163/164 (Supplementary Table 3). Successful replacement of *glgX* on cosmid PI1_F8 with the apramycin resistance cassette was confirmed by PCR

using primer pair MW165/166. The mutagenized cosmid was subsequently transformed into *E. coli* ET12567/pUZ8002 for conjugation in *S. venezuelae*. Conjugation was carried out overnight at rt on SFM plates. The following day, plates were overlayed with 250 μg/ml nalidixic acid and 500 μg/ml Apr in 2 ml ddH$_2$O and incubated at 30 °C until colonies appeared. The desired mutant arising from a double crossing over event was screened for a Kan sensitive and Apr resistant phenotype and confirmed by PCR using primer pair MW165/166.

### Construction of *S. venezuelae* strains overexpressing *glgX* and expressing *glgX*-FLAG for complementation and Western blotting

The integrative plasmids pIJ10257_*glgX* and p3xFLAG_*glgX* were transformed into *E. coli* ET12567/pUZ8002 for conjugation into *S. venezuelae* on SFM agar. The plates were incubated overnight at rt before overlaying with 2 ml ddH$_2$O containing 250 μg/ml nalidixic acid and 500 μg/ml Hyg and incubated at 30 °C until colonies appeared. Exconjugants were subsequently re-streaked twice on plates containing nalidixic acid and Hyg.

### c-di-GMP extraction and quantification

For extraction and quantification[52] of c-di-GMP, at the indicated time points, 5 ml samples for nucleotide extraction and 1 ml samples for protein quantification, respectively, were taken from *S. venezuelae* grown in MYM. For c-di-GMP extraction, cell pellets were suspended in 800 μl Extraction mixture II (acetonitrile/methanol/water [2:2:1]), shock frozen for 15 s in liquid nitrogen and heated for 10 min at 95 °C. After cooling on ice, samples were disrupted using the BeadBlaster at 4 °C with 2 cycles at 6 m/s for 45 s and 2 min interval. After centrifugation, supernatants were transferred into 2 ml reaction tubes. Remaining pellets were suspended in 600 μl Extraction mixture I (acetonitrile/methanol [1:1]), pulsed two times for 30 s at 6 m/s with a 60 s interval, incubated on ice and centrifuged as above. The extraction with 600 μl Extraction mixture I was repeated once. All supernatants (~2 ml) were combined and stored for protein precipitation for two days at −20 °C. Precipitated proteins were removed by centrifugation and the precipitation step was repeated. Finally, samples were air dried in a SpeedVac Plus SC110A connected to Refrigerated Vapor Trap RVT100 (Thermo Scientific) at low temperature settings and analyzed using LC-MS/MS[53].

Samples for protein quantification were suspended in 800 μl 0.1 M NaOH, transferred into 2 ml screw cap tubes prefilled with 0.1 mm silica beads (Biozym) and heated for 10 min at 98 °C. Cell lysis was performed in BeatBlaster with 2 pulses for 30 s at 6 m/s and an interval of 2 min. After centrifugation at 4 °C, the supernatant was saved and the extraction step was repeated. Supernatants were combined and protein concentration was determined via Bradford using Roti-Quant. For normalization of c-di-GMP concentration to the protein amount, the following formula was used:

$$\frac{c-di-GMP[nM] \times 200}{cV[ml] \times c590[\frac{\mu g}{ml\,cells}]} = \frac{c-di-GMP[pmol]}{protein[mg]} \qquad (1)$$

### Quantification of glycogen

Spores of the respective *S. venezuelae* strains were spread on MYM plates and grown (as a lawn of cells) for 5 days at 30 °C. Approximately 300 mg of cells were collected, shock frozen in liquid nitrogen and stored at −80 °C until further use. After resuspension in 500 μl PBS, cells were disrupted using the BeadBlaster (Biozym) and 0.1 mm glass beads at 4 °C (5 cycles at 5.5 m/s for 30 s with a 60 s interval). Samples were centrifuged for 20 min at 4 °C and 35,000 × *g* and the supernatants were transferred to new tubes, this step was repeated twice. Protein concentration was determined by the Bradford assay using Roti-Nanoquant (Roth). For glycogen precipitation experiments[54],

200 μl of the supernatant was mixed with KOH to a final concentration of 30% (w/v) and incubated at 95 °C for 2 h. After the addition of ice-cold ethanol to a final concentration of 70% (v/v) the glycogen was precipitated at −20 °C for 20 h. After centrifugation at 4 °C and 35,000 × *g* for 15 min the pellet was washed with 70% (v/v) ethanol followed by 5 washing steps with 98% (v/v) ethanol. The pellet was dried using a speed-vac for 20 min at 60 °C and finally resuspended in 200 μl PBS by shaking at rt for 2 h at 600 rpm. For glycogen quantification, the MAK016 assay kit (Sigma-Aldrich) was used. 25 μl of appropriate dilutions of the samples were mixed with hydrolysis buffer and further processed according to the manufacturer's instruction.

### Western blotting

For the detection of FLAG-tagged GlgX, 5–10 ml culture of the respective *S. venezuelae* strain were taken after 8-22 h of growth. *S. venezuelae* WT served as negative control and was harvested after 22 h of growth. Culture samples were centrifuged at 3000 × *g* for 30 min and the bacterial pellet was suspended in 1 ml lysis buffer (50 mM Tris pH 7.5, 150 mM NaCl, 5% (v/v) glycerol) containing protease inhibitors (Roche). Bacteria were lysed by bead beating (Biozym; 5 cycles at 6.00 m/s; 45 s pulse; 2 min interval) and subsequently centrifuged at 35,000 × *g* at 4 °C for 10 min to remove cell debris. Total protein concentration was determined using Bradford assay (Roth) and the cell lysate was then diluted to 1 μg total protein/μl in loading buffer. Samples were boiled for 10 min, centrifuged for 5 min at 35,000 × g and 20 μl aliquots (=20 μg total protein) were separated on a 10% SDS-PAGE gel containing 0.5% (v/v) trichloroethanol. Following electrophoreses, proteins were UV cross-linked for 5 min and subsequently visualized under UV light to insure equal sample loading. Next, proteins were transferred to a polyvinylidene difluoride (PVDF, Roth) membrane by electroblotting using a semi-dry blotting system from Bio-Rad. For the detection of FLAG-tagged proteins, the anti-FLAG antibody (Sigma) and the HRP-conjugated anti-mouse (GE Healthcare) were used at 1:10,000 and 1:20,000 dilutions, respectively. Antibody incubations were performed in Tris-buffered saline pH 7.5, 0.1% (v/v) Tween 20 (TBST) buffer containing 5% (v/v) milk. Blots were developed using Clarity™ Western ECL Substrate (BioRad) and a ECL Chemocam Imager (Intas Pharmaceuticals Limited).

### Bicinchoninic acid assay

Hydrolysis of carbohydrates was assessed in vitro by measuring the accumulation of reducing sugar ends using the BCA assay[42] and D-glucose as a standard for calibration (Supplementary Fig. 11). Purified enzyme was diluted to 1 mM in 20 mM (*N*-2-hydroxyethylpiperazine-*N*′−2-ethanesulfonic acid) HEPES (pH 7.5; ionic strength, 160 mM [adjusted with NaCl]) containing 10 mM MgCl$_2$ and 50 mM c-di-GMP (BioLog). Samples were incubated at rt for 5 min and reactions were initiated by the addition of 0.05% of the relevant carbohydrate provided by Sigma-Aldrich: glycogen from rabbit (cat. no G8876); amylose from potato (cat. no A0512); amylopectin from maize (cat. no. 10120); and pullulan (P4516). The mixture was incubated at 37 °C in a heat block for 1 h. Next, an equal volume of BCA reagent solution (250 mM Na$_2$CO$_3$; 144 mM NaHCO$_3$; 2.5 mM sodium bicinconinic acid (Sigma); 6 mM L-serine (Roth); 2.5 mM CuSO$_4$ x 5 H$_2$O (Roth)) was added, reactions were vortexed vigorously and incubated at 80 °C in a heat block for 30 min. Assay mixtures were then cooled to rt. 125 μl sample aliquots were transferred in duplicates to a 96 well plate and the absorbance at 562 nm was measured using a plate reader.

### Differential radial capillary of ligand assay

For DRaCALA assays, purified His$_6$-GlgX (15 μg) was mixed with 0.5 μl undiluted [$^{32}$P]-c-di-GMP (42 nM) (Hartmann Analytic GmbH, Braunschweig) in binding buffer (25 mM Tris pH 8; 150 mM NaCl; 2.5% (v/v) glycerol; 5 mM MgCl$_2$) and the reaction was incubated at rt for 5 min. For competition assays, 100 μM nonlabeled nucleotides were

added to the samples and incubated at rt for 5 min prior to the addition of [$^{32}$P]-c-di-GMP. Reactions (10 μl) were spotted on the same nitrocellulose membrane (Roth), air dried for 10 min and analyzed via phosphorimaging.

## Microscale thermophoresis

For MST experiments, purified GlgX was labeled using Monolith NT protein labeling kit RED-NHS (amine reactive) dye (NanoTemper Technologies GmbH) according to manufacturer´s guidelines. The protein concentration was adjusted to 20 μM in labeling buffer and the labeling reaction was carried out at rt for 30 min in the dark. Excess fluorescence dye was subsequently removed from the labeled protein by gel filtration using a dye removal column provided by the supplier and equilibrated with MST buffer (25 mM Tris pH 8, 150 mM NaCl, 2.5% (v/v) glycerol, 5 mM MgCl$_2$, 0.05% (v/v) Tween). A UV/VIS spectrophotometry at 650 and 280 nm was used to determine the degree of labeling. Next, labeled GlgX was adjusted to 80 nM in MST buffer and a series of 16 1:2 dilutions of c-di-GMP (BioLog) were prepared using the same buffer. An equal volume of protein and c-di-GMP dilutions were mixed resulting in 16 samples with 40 nM protein each and c-di-GMP concentrations ranging from 2 mM to 0.00146 mM. Samples were then loaded into Monolith NT.115 Premium Capillaries (NanoTemper Technologies) and MST was measured using a Monolith NT.115 instrument (NanoTemper Technologies). Instrument parameters were set to 20% LED power and 40% MST power and the data were analyzed using MO.Affinity Analysis software version 2.3; NanoTemper Technologies.

## Label-free thermal shift assays

For label-free thermal shift assay experiments, 10 μM purified protein in 25 mM Tris pH 8, 150 mM NaCl, 2.5% (v/v) glycerol, and 5 mM MgCl$_2$ was incubated without or with (1 mM) of nonlabeled c-di-GMP or c-di-AMP, respectively, at rt for 10 min. Reactions were subsequently analyzed in a Tycho NT.6 (NanoTemper Technologies) device with a 30 K/min thermal ramp and the internal fluorescence at 330 and 350 nm was recorded. Data analysis and calculation of derivatives was done using the internal evaluation features of the Tycho instrument.

## Transmission electron microscopy

For transmission electron microscopy analysis, single colonies of *S. venezuelae* were cut out of an agar plate and fixed in a solution of 2.5% (v/v) glutaraldehyde in 0.05 M sodium cacodylate, pH 7.3. Using a Leica EM TP machine (Leica Microsystems), the samples were washed in 0.05 M sodium cacodylate and post-fixed with 1% (w/v) OsO$_4$ in 0.05 M sodium cacodylate for 60 min at rt. After washing and dehydration with ethanol, the samples were gradually infiltrated with LR White resin (London Resin Company) according to the manufacturer's instructions. The polymerized material was sectioned with a diamond knife using a Leica EM UC6 ultramicrotome (Leica Microsystems). Sections of ~90 nm were picked up on 200-mesh gold grids that had been coated in pyroxylin and carbon. The grids were stained for α-glucans by placing them in 1% (v/v) periodic acid for 20 min at rt. Then, they were washed in water, placed in 0.2% (w/v) thiocarbohydrazide in 20% (v/v) acetic acid overnight, washed in acetic acid then water, and finally stained with 1% (w/v) silver proteinate for 30 min in the dark. The grids were dried and viewed in a Thermo Fisher Talos F200C transmission electron microscope at 200 kV and imaged using a Gatan OneView camera (AMETEK (GB) Limited) running the DigitalMicrograph (version 3.5) software.

## *S. venezuelae* GlgX expression and purification

For structural and biochemical studies the gene encoding *S. venezuelae* GlgX and GlgX mutants, GlgX$_{E47A-R49A}$ and GlgX$_{R438A-R579A}$, were codon optimized for *E. coli* expression, purchased from Genscript Corporation and subcloned into pET15b such that a His-tag was expressed on

the proteins for purification (Piscataway, NJ, USA; http://www.genscript.com). C41(DE3) cells were transformed with these expression vectors. Cells with each expression construct were grown at 37 °C in LB medium with 0.17 mg/ml ampicillin to an OD$_{600}$ of 0.5 then induced with 0.5 mM isopropyl β-d-thiogalactopyranoside (IPTG). The induction was done overnight at 15 °C. The cells were harvested by centrifugation and then resuspended in Buffer A (50 mM Tris-Cl pH 7.5, 300 mM NaCl, 5% (v/v) glycerol, 5 mM MgCl$_2$, 0.5 mM β-mercaptoethanol (βME)), with 1X protease inhibitor cocktail. The resuspended cells were then disrupted with a microfluidizer and cell debris was removed by centrifugation (35,000 × g, 4 °C, 60 min). The supernatant was loaded onto a cobalt NTA column. The column was washed with 500 ml of 5 mM imidazole in buffer A. The elution of the proteins were done with the following imidazole concentrations (30 mM, 35 mM, 40 mM, 45 mM, 50 mM, 55 mM, 60 mM, 65 mM, 80 mM, 100 mM, 200 mM, 300 mM, 500 mM, 1000 mM) in Buffer A. Fractions were analyzed by SDS-PAGE and those containing the protein were combined. His-tags were cleaved by thrombin digestion overnight at 37 °C using a thrombin cleavage capture kit (Novagen). The cleaved His-tags were removed by loading the cleavage reaction onto a Ni-NTA column and collecting the flow through. The proteins were concentrated using centricons with a 50 kDa MW cutoff.

For DRaCALA, nanoDSF and BCA assays, *glgX* variants were overexpressed from the pET15b vector in *E. coli* Rosetta (DE3). Gene expression was induced by the addition of 1 mM IPTG at OD$_{578}$ of 0.5. Cultures were shifted from 37 °C to 16 °C and incubated overnight with agitation. After centrifugation at 4000×*g* for 15 min the bacterial pellet was suspended in buffer B (50 mM Tris pH 7.5, 150 mM NaCl, 5% (v/v) glycerol) containing protease inhibitors (Roche). Bacteria were lysed by 3–4 passages through a french press cell and subsequently centrifuged at 17000 rmp for 40 min. The lysate was then applied to 1 ml equilibrated 50% Ni-NTA SuperFlow (iba) and incubated overnight at 4 °C. The following day, the matrix was washed with buffer B, and then with buffer B containing 10 mM imidazole followed by a second wash with buffer B containing 50 mM imidazole. Proteins were eluted with 5 × 1 ml of buffer B supplemented with 250 mM imidazole. Fractions containing protein were pooled and dialyzed twice against 1.5 liters of a buffer consisting of 25 mM Tris pH 8.0, 150 mM NaCl, 2.5% (v/v) glycerol and 5 mM MgCl$_2$ at 4 °C with stirring. Proteins were aliquoted, flash frozen in liquid nitrogen and stored at −80 °C.

## Crystallization and data collection of *S. venezuelae* apo GlgX, GlgX-c-di-GMP, GlgX-c-di-GMP-acarbose complexes

For crystallization, tag-free and His-tagged *S. venezuelae* WT GlgX was concentrated to 15 mg/ml. Wizard screens I to IV and cryo screens I and II were used for screening at rt by the hanging drop vapor diffusion method. Crystals of the His-tagged apo GlgX were produced by mixing the protein (at 15 mg/ml) 1:1 with a solution consisting of 9% (v/v) propanol, 675 mM ammonium citrate/ammonium hydroxide pH 8.5. The same crystals were grown using the His-tag free protein. These crystals were cryopreserved by dipping them in a solution consisting of the crystallization reagent supplemented with 20% (v/v) glycerol before plunging into liquid nitrogen. Crystals of GlgX bound to c-di-GMP were generated by mixing a GlgX solution at 15 mg/ml with 3 mM c-di-GMP with a crystallization reagent composed of 40% (v/v) ethylene glycol, 0.1 M sodium acetate pH 4.6. The crystals could be cryopreserved straight from the drop. The GlgX-c-di-GMP-acarbose complex was produced by mixing GlgX (10 mg/ml) with c-di-GMP and acarbose at final concentrations of 3 mM and 10 mM, respectively. Two crystal forms were generated. One crystal form grew when mixing the solution 1:1 with a reagent containing 50% (v/v) MPD, 0.1 M Tris pH 8.5 and 200 mM ammonium phosphate monobasic. The second crystal form was produced by mixing the GlgX-c-di-GMP-acarbose complex 1:1 with a crystallization solution composed of 30% (v/v) PEG 300, 100 mM NaCl and 0.1 sodium acetate pH 4.6 Both GlgX-c-di-GMP-

acarbose crystals could be cryo-preserved straight from the drop. X-ray intensity data for the crystals were collected at ALS beamline 5.0.2 and 5.0.1 processed with XDS (Supplementary Table 1).

### Structure determination of *S. venezuelae* apo GlgX, GlgX-c-di-GMP, and GlgX-c-di-GMP-acarbose complexes

The crystal structure of the apo GlgX complex was phased/solved by molecular replacement (MR) using one subunit of the TreX structure, 2VUY (https://www.rcsb.org/structure/2vuy), as a search model. There were twelve subunits in the crystallographic asymmetric unit (ASU)[55] but only eight were identified in MR. After one round of refinement in Phenix (version 1.19)[56] a subunit of the refined model was used to identify the additional four GlgX subunits. The structure was refined further in Phenix and after three rounds of refinement, the side chains for the *S. venezuelae* protein were then substituted for the TreX residues and regions that were clearly different were rebuilt. Despite the resolution, the electron density map was of excellent quality and revealed clear density for most of the side chains. After multiple rounds of extensive rebuilding in Coot[55], the structure was refined in Phenix[56] to convergence. The structure of the GlgX-c-di-GMP was then solved by MR using one of the *S. venezuelae* apo GlgX subunits as a search model. There are two GlgX dimers in the ASU. Density was clear for the c-di-GMP molecules after the first round of refinement. In addition, several regions near the bound c-di-GMP and several active site regions showed marked differences from the apo structure and had to be rebuilt in Coot[55]. After several rounds of refinement, the $R_{work}/R_{free}$ values converged to 20.5%/27.5% to 3.34 Å resolution. The structure of the GlgX-c-di-GMP-acarbose (pH 8.5) structure was solved by MR (Phenix)[56] using a subunit of the c-di-GMP bound *S. venezuelae* GlgX as a search model. There were four GlgX dimers in the ASU, which were found via MR. After three cycles of Phenix_refine[56] there was clear density for the c-di-GMP molecules bound in the same location as observed in the GlgX-c-di-GMP structure as well as density for three sugars of the acarbose bound in each GlgX active site. After adding the c-di-GMP and acarbose molecules, refinement was carried out to final $R_{work}/R_{free}$ values of 23.6%/30.3% to 3.6 Å resolution. The structure of the GlgX-c-di-GMP-acarbose complex crystals that grew at pH 4.6 was solved by MR using the GlgX from the GlgX-c-di-GMP structure as a search model. There were two GlgX dimers in the ASU. Unlike the GlgX-c-di-GMP-acarbose structure that was crystallized from a solution at pH 8.5, in this structure there was density for all four sugars of the acarbose. Density for the c-di-GMP molecules were found in the same location as the prior structures. After fitting the c-di-GMP and acarbose molecules and adding solvent, the model was refined to convergence. See Supplementary Table 1 for data collection and final refinement statistics.

### Fluorescence polarization binding experiments

To measure c-di-GMP binding to GlgX and GlgX mutants, 2′-O-(6-[Fluoresceinyl]aminohexylcarbamoyl)-cyclic diguanosine monophosphate (2′-Fluo-AHC-c-di-GMP) (BioLog), was used as a fluoresceinated reporter ligand. This molecule is conjugated via a nine atom spacer to one of the c-di-GMP 2′ hydroxyl groups. This ligand was chosen as the GlgX-c-di-GMP structure revealed that one ribose hydroxyl of the GlgX-bound c-di-GMP is solvent exposed. 2′-O-(6-[Fluoresceinyl]aminohexylcarbamoyl)-cyclic diadenosine monophosphate (2′-Fluo-AHC-c-di-AMP) (BioLog) was also used in binding studies to assess specificity of GlgX for c-di-GMP. The experiments were performed in a buffer consisting of 25 mM Tris-HCl pH 7.5, 150 mM NaCl and 5 mM MgCl₂, which contained 1 nM 2′-Fluo-AHC-c-di-GMP or 2′-Fluo-AHC-c-di-AMP at 25 °C. Increasing concentrations of GlgX or GlgX mutant were titrated into the reaction mixture to obtain their respective binding isotherms. The resultant data were plotted using KaleidaGraph (version 4.5, serial # 8011073 (synergy software)) and the curve fit to deduce binding

affinities. All the experiments were performed in technical triplicates and the error (SD) between measurements noted.

### Bioinformatic analyses

The Patric command line (patricbrc.org (https://www.patricbrc.org)) was employed to download all protein sequences from complete genomes that belong the PGF_05732875 family. Most of the members of this family are annotated as "limit dextrin alpha-1,6-maltotetraoose-hydrolase". The total number of downloaded sequences was 13,585 proteins. The protein lengths ranged from ~600 to ~900 amino acids. Accordingly, we removed outlier proteins that were either <500 or >950 amino acids, resulting in 12,955 proteins. We performed Clustal Omega alignment and searched for our reference *S. venezuelae* protein with the unique Patric identifier fig|54571.16.peg.5982. Using the alignment, proteins containing the RxxD motif involved in c-di-GMP binding were identified which led to the selection of 280 proteins (Supplementary Table 2). We used WebLogo 3 (http://weblogo.threeplusone.com/create.cgi)[57] to visualize the conservation of c-di-GMP-binding residues in GlgX homologs. For that, protein sequences of the GlgX homologous proteins were compiled in a fasta file. We used Clustal Omega[58] to align the amino acids.

### Reporting summary

Further information on research design is available in the Nature Research Reporting Summary linked to this article.

## Data availability

All data generated or analyzed during this study are included in the published article or can be obtained from the Protein Data Bank. Coordinates and structure factor amplitudes have been deposited with the Protein Data Bank under the accession codes 7U3A, 7US9, 7U3B and 7U3D. The Patric database (patricbrc.org (https://www.patricbrc.org)) was employed to download all protein sequences from complete genomes that belong the PGF_05732875 family. Source data are provided with this paper (see Source_data_file).

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

## Acknowledgements

Research in Natalia Tschowri's lab is funded by the DFG Emmy Noether-Program (TS 325/1-1) and the DFG Priority Program SPP 1879 (TS 325/2-2). We thank Stephen Bornemann for providing strains, Julia Schneider for technical assistance and the research core unit metabolomics at the Hannover Medical School for support with LC-MS/MS analysis. We acknowledge Alison M. Smith, Karl Forchhammer, and Niels Neumann for scientific consultation. We thank Kim Findlay of the JIC Bioimaging facility for assisting with the electron microscopy and Neil Holmes for handling the samples. The research in the Schumacher lab was supported by National Institutes of Health grants (R35GM130290). We acknowledge beamline 5.0.2 and 5.0.1 for X-ray diffraction data collection. The ALS (Berkeley, CA) is a national user facility operated by Lawrence Berkeley National Laboratory on behalf of the US Department of Energy under Contract DE-AC02-05CH11231, Office of Basic Energy Sciences. Beamline 5.0.2 and 5.0.1 of the ALS, a US Department of Energy Office of Science User Facility under Contract DE-AC02-05CH11231, is supported in part by the ALS-ENABLE program funded by the NIH, National Institute of General Medical Sciences, Grant P30 GM124169-01.

## Author contributions

N.T. and M.A.S. designed the study. All authors designed and interpreted experiments, which were performed by M.A.S., M.E.W., M.H., R.S., M.M.AB., K.S.S., E.B., K.G., A.L., and N.T. The figures were made by M.E.W, M.A.S., M.M.AB., A.L. and N.T. The paper was written by M.A.S. and N.T. with input from the other authors.

## Competing interests

The authors declare no competing interests.
