## [Peer Review File · Nature Communications]

Allosteric regulation of glycogen breakdown by the second messenger cyclic di-GMPReviewer #1 (Remarks to the Author):

Attachment on the following page.

Review of Schumacher et al. 2022 (Nature Communications)

The authors present work of high significance and impact for the field by studying the function of cyclic di-GMP. c-di-GMP fulfills a global regulatory role in diverse bacterial functions, including biofilm formation, motility, virulence and cell cycle progression by binding to diverse protein effectors and riboswitches. In this manuscript the authors elucidate a novel allosteric regulatory function of the second messenger by characterizing its interaction with GlgX, the glycogen debranching enzyme in *S. venezuelae*. The authors verify c-di-GMP binding to GlgX, identify the binding site and later characterize a new binding motif. They establish c-di-GMP-mediated stimulation of GlgX activity and identify active site residues, which when mutated ablate GlgX activity. Beside a relatively weak attempt at analyzing GlgX substrate specificity, the authors demonstrate a key role of GlgX in *S. venezuelae* glycogen homeostasis by comparing strains expressing WT-levels of GlgX with those lacking or overexpressing GlgX. Finally, they solve multiple structures of GlgX and find support for c-di-GMP-mediated conformational change from inactive to active form of the enzyme. While the scientific approach and experimental seems generally appropriate, the work at times lacks in thoroughness necessary to make solid conclusions. Additional data is necessary to bolster some conclusions. The authors should include more details in the method section to improve the reproducibility of their work. Detailed concerns are found below.

Major concerns:

Section starting line 120: It is not described how the stages vegetative growth (8 to 10 h), transition to sporulation (12 to 16 h) and sporulation (18 to 20 h) are confirmed? How do the authors know in which stage the culture is? To improve reproducibility it should be explained in more detail what time point the authors designate as time point zero. The conclusion of a 'rise of c-di-GMP during spore formation' is only valid if the authors confirm that the bacteria are in the assumed stages. In the Methods part should explain how the liquid culture of *S. venezuelae* is maintained, how cells are synchronized, when is "the start of the culture", and what are the conditions at this time point (cell density etc.)? In the results it should be mentioned how cell density changes over the experimental growth period.

Section starting line 144: The authors generate two transgenic lines with two GlgX alleles (Vnz25270 and Vnz29170) expressed under the native promoter. Protein presence is tested by Flag Western-blot. The absence of expression of one of the Flag-GlgX proteins is interpreted as absence of the native form of this GlgX protein in the bacteria. However, this is only true under the assumption that the generation of both transgenic lines was equally successful. The authors should show mRNA expression levels of the transgenic/flag-tagged as well as of the WT alleles.

Figure 1C and Supplemental Figure S2: The authors compare binding of ligands to two purified proteins (Vnz25270 and Vnz29170) using DRaCALA assays. They draw conclusions about both proteins from an n of 1 experiment. The DRaCALA assays enable quantitative analyses. The authors should augment their data with appropriate technical replicates, and conduct quantitative analyses. The authors should also characterize the concentration and amount of radiolabel in the [32P]-c-di-GMP preparation they are using for their assay. The DRaCALA assays displayed in Figure S2 look unusual as the common outer circle is not visible, indicating changes in ligand concentration or radiolabel. To draw conclusions about ligand binding to both proteins, the authors should show purity analyses of both proteins. The authors do spot equal amounts of protein, but if the in the one case purification was not as effective, less of the GlgX paralogue will be spotted on the membrane, resulting in much less ligand retention in the center. Also, the authors should indicate which spots were developed imaged on the same membrane (for both figures) and can therefore be directly compared.

Section starting line 179: The authors show maximal activation of GlgX by c-di-GMP at 50uM. However, they do not show that their reducing ends assay still responds to increased amounts of reducing ends. The assay should be run with D-glucose as a calibration standard for reducing ends and data should be expressed as 'Reducing ends [xmol glucose equivalents]'.

In addition the stimulation should be presented in a scatter plot to better visualize the dose-response-relationship.

Regarding Fig. 2B: The authors should provide details regarding the origin of the glycogen, pullulan, amylopectin (vendor, catalogue number). The comparison of GlgX action on different carbohydrate substrates seems to be skewed by the use of much less carbohydrate starting material (significantly smaller signal at TLC origin for amylopectin and amylose). The authors should note that commercial preparations of polyglucans are of very different purity, equal carbohydrate concentration should therefore be based on glucose residues, not just based on weight dry material per volume. The TLC assay should be complemented with reducing end quantification, ideally including a quantification of branch-points per glucose unit for each carbohydrate substrate.

Line 196-198: The authors compare debranching efficiency a highly branched substrate (glycogen) with an essentially unbranched substrate (amylose) on the grounds of the amount of released chains. In the case of amylose the reaction is much less likely to happen in the first place, simply because they are just much lower in abundance. Why do the authors deduce lower GlgX efficiency from a lesser release of chains from amylose?

Thin layer chromatography details (e.g. plate type, spotting volume) are missing in the method section. The authors should explain what they think the signals away from the TCL origin signify in Fig. 2B. I assume it is glucan chains released by GlgX, with different mobility for different chain lengths. Should not debranching of pullulan lead to chain lengths of multiples of three (maltotriose * n)? However, the pattern of the released chains is identical to that released from glycogen. The authors have to explain that at least in the method part. Also, is it possible that chains were released from amylopectin, but they were too immobile to separate from the origin because as the authors state the amylopectin chains are much longer than those in glycogen?

The authors should change all their bar graphs to include individual data points and indicate the error bar definition in each figure legend.

The effects of glgX knockout and overexpression on Streptomyces development could have an effect on glycogen levels. The authors state they did not see an effect of glgX deletion on development. They should explain and show the evidence for that. Also, the authors deduce from the absence of green spore pigment formation in glgX overexpressing cells that sporulation is delayed. To make this statement the authors must show time-dependent onset of sporulation. If they do not show that sporulation in overexpressing cells is still happening (but later than in WT) this is only speculation.

Figs 4 and 5: There is no reference in the text to figure 4B. 5A and 5C are referred to after introducing fig.6, which makes the whole thing a bit confusing. 5C could be moved to figure 1 and discussed earlier. There is a reference to 5B (line 236), but it doesn't seem to be showing what is referred to in the text. In line 239 there is a reference to fig 5D, which does not exist. The description for Figure 5B is missing in the figure legend. The authors should consolidate these things.

The crosslinking experiment in Fig. S5 is not explained in sufficient detail. Was this done under physiological protein concentrations? How was this ensured? How did the authors verify that high MW bands are multimers of glgX? The lower DETR mutant gel shows much lower band intensities, which indicates lower sensitivity of the signal detection. This could lead to an underestimation of the high MW bands. When I adjust the gels to similar intensity of the low MW bands (see figure below), it becomes apparent that crosslinking is very similar in the mutant protein, however not d-c-GMP dependent but only DSS dependent. The authors should discuss this. The authors use different size markers in both gels. Can the authors please indicate the commercial source?

Bottom gel with adjusted contrast.

Minor concerns:

Line 48: Since there is evidence that mammalian glycogen contains non-negligible amounts of glucosamine residues, the sentence should be changed to 'Glycogen is a polysaccharide largely consisting of α -1,4-linked glucose subunits with α -1,6-linked glucose at the branching points.'

Line 50: It should be made clear that only liver glycogen (not skeletal muscle glycogen) is considered to be used to replenish the blood with glucose. Nevertheless the muscle is playing a role in maintaining blood glucose levels by insulin-dependent glucose uptake and glycogen synthesis stimulation. There is a 2021 chapter on mammalian glycogen metabolism from Bart Pederson which summarizes the roles of glycogen in different organs quite nicely.

Line 56: Not sure it makes sense to say "prolonged survival rates". The survival is longer, the rates are higher... Are there such things as longer rates?

Line 68: Since this manuscript focuses on GlgX, it would be nice to characterize the debranching reaction bit more in detail. E.g. is the bacterial debranching reaction considered to be direct or indirect?

Line 147: The authors should include a reference to fig. S3 here.

Line 156: The authors should clarify that they refer to Vnz29170 GlgX protein when they first speak about His6-tagged protein for nanoDSF and DRaCALA assays.

Section starting line 168: The current data doesn't rule out that c-di-GMP is still binding but not stabilizing the protein anymore. It is possible that the motifs are not necessary for binding but for stabilization. The authors should include the mutated GlgX proteins in their DraCALA analyses. Regarding Figure S4, the original RxxD motifs seem to be misspelled (i.e. RETR, and RLVR). Later in the manuscript the authors describe fluorescence polarization binding experiments that verify the binding data in Fig. 1. I suggest to move the FP data into figure 1 to substantiate the binding data when binding is discussed at first.

Line 190: Amylose should be called 'essentially unbranched chain'. The amylopectin description seems a bit peculiar. The authors should add a citation to identify the source of their information.

Fig. S3: The authors should indicate in the figure legend what '.*' symbols signify.

Reviewer #2 (Remarks to the Author):

In this recent work the Tschowri/Schumacher team uncovers the regulation and molecular mechanism of glycogen storage control by the second messenger c-di-GMP in *Streptomyces*, an organism that is a major source of antibiotics involving a complex developmental program. Although c-di-GMP has been linked to several pathways pertaining to biofilm formation, motility, virulence and cell cycle regulation, this report is the first in-depth study into the second messenger's role in regulating glycogen metabolism, representing a truly novel finding that should be of great interest to the field.

Previously c-di-GMP has been established as a major regulator for the transition from a vegetative to spore forming state of *Streptomyces* due to its ability to bind to and regulate effector proteins BldD and WhiG, central regulators for distinct steps in the pathway. Cellular c-di-GMP levels are high during the vegetative growth phase but drop as the developmental transition proceeds. Here, the authors show that there is a second phase where c-di-GMP levels rise, namely as cells sporulate. Using an unbiased affinity enrichment for c-di-GMP-binding proteins, the authors identify the glycogen-debranching enzyme GlgX as a specific interactor for the second messenger. GlgX enzyme activity is stimulated in the presence of c-di-GMP and the enzyme is crucial for glycogen breakdown in *Streptomyces*. Crystal structures of *Streptomyces* GlgX in its apo-form, bound to c-di-GMP and bound to c-di-GMP and a substrate analog reveal the molecular mechanism regarding the activation of GlgX by the dinucleotide. Specifically, the apo-form is monomeric and displays an active site in a catalytically incompetent state. C-di-GMP binding, via a long-range, allosteric effect, reorders the active site, with the enzyme adopting an active state that can accommodate substrate. The active state appears to involve a weak GlgX dimer, although the functional relevance of this detail was not thoroughly tested here. Notably, the specific interaction of c-di-GMP with GlgX expands the repertoire of reported cyclic dinucleotide binding sites.

The experiments presented in the manuscript contain important controls, including sequence- and structure-informed mutations, linking the structural insight to the regulation of GlgX in cells. The flow from identifying a second phase of c-di-GMP during *Streptomyces* spore formation to the identification of a crucial metabolic enzyme to its structure in multiple functional snapshots depicting the mode of c-di-GMP activation of GlgX renders this report complete with several important, novel discoveries on the way.

The only major point that I feel was not fully addressed pertains to the dimerization of GlgX and its relevance for the activation of the enzyme by c-di-GMP:

The crosslinking data in Figure S5 are not fully convincing in showing dimerization of GlgX in response to c-di-GMP. This is in line with a modest, yet potentially sufficient dimerization interface observed in crystals, but its impact on the enzymatic mechanism was not tested. Have the authors tried other methods for reading out oligomerization? Gel filtration (ideally coupled with a multi-light scattering setup), analytical ultracentrifugation or mass photometry could yield more quantitative data.

Along the same lines: While the authors have done a great job validating the structural insight by using site-directed mutants designed based on the structural models, such an analysis was omitted for the dimerization interface. This leaves open a mechanistic question: Is dimerization required for GlgX activity? Would it be possible to design a mutant variant of GlgX that would be predicted to still bind c-di-GMP and substrate but that would be impaired in dimerization? Or would a dimerization mutant be defective in c-di-GMP and/or substrate binding?

Minor points:

1. The supplemental movie is effective in presenting the overall structure and the mode of c-di-GMP binding. Have the authors considered illustrating the conformational changes from the apo to the c-di-GMP-bound state? This could supplement the still images shown in the figures.

2. Figure 7: This figure shows a superposition of the c-di-GMP bound state with a substrate analog- and c-di-GMP-bound conformation. It would be informative to also

show a close-up view of the substrate analog-bound active site with that of the apo-GlgX structure superimposed. This would highlight the structural changes upon activation by c-di-GMP relative to the substrate, adding clarity to the mechanism of activation.

3. Have the authors attempted to crystallize a c-di-GMP-free GlgX bound to the substrate analog? Along those lines, does substrate or substrate analog alter the oligomeric state of GlgX in the presence or absence of c-di-GMP?

Reviewer #3 (Remarks to the Author):

Schumacher et al. 2022. Nat Comm.

Schumacher et al. report about the regulation of a glycogen-debranching enzyme (GlgX) from a *Streptomyces* species by the second messenger c-di-GMP. They show direct binding of c-di-GMP and saturable activation of the enzymatic activity by the compound. Several crystal structures are presented which reveal the c-di-GMP binding site, which is validated by mutagenesis, and the structural changes induced by c-di-GMP binding. The study has been performed competently and the results are presented well. GlpX is thus one of the few c-di-GMP targets that have been studied thoroughly both biochemically and structurally. However, it is expected that there are many more unrelated c-di-GMP binding proteins, such that the results of the current study cannot be generalized, in particular with respect to the mode of c-di-GMP binding and the induced allosteric change. Therefore, I think that this paper would be more suited to be published in a specialized journal.

Below, I briefly list a few points that occurred to me upon first reading:

* Why is c-di-GMP sensed only in the late developmental stage (l. 140)? This is mentioned again in the Discussion (l. 369). I guess it would be due to the higher K_d for GlpX binding (compared to the other c-di-GMP binders mentioned). Would be appropriate to elaborate on this. Also discuss the effect of high c-di-GMP levels on BldD, etc.

* Binding of c-di-GMP to GlpX has been studied by several techniques (paragraph starting at l. 138), but the quantitative analysis by fluorescence polarisation measurement is reported only much later (l. 270ff). Latter study employed FLUO-c-di-GMP ($K_d=8 \mu\text{M}$). To get the affinity of unmodified c-di-GMP a corresponding competition experiment is missing.

line 29: GlgX-c-di-GMP and apo GlgX structures reveal >> comparison of the GlgX-c-di-GMP and apo GlgX structures reveals

line 176: "that the RxxD motif at position 49-52 is essential for ligand binding". The conclusion reaches too far. The swap experiment reveals that the R, the D, or both are essential for binding.

line 186: "maximal enzymatic activity in the presence of 50 μM c-di-GMP" >> activity is saturating at 50 μM c-di-GMP. By proper quantitative evaluation of the activation curve (probably requiring further experimental data) it should be possible to get the activation constant or EC_{50} of the c-di-GMP activator. Would be interesting to compare this with the binding data.

line 227: mention phasing method

line 236: Fig. 5B >> Fig. 4B

line 236: change no >>> There are no protein-protein interfaces in the crystal that bury

a significant amount of buried surface area

line 267: "showing that the RETD49-52 motif is essential for c-di-GMP binding. " Up to here only the involvement of R49 has been shown.^[1]_{SEP}

line 277: Our analyses showed >> As shown above (Fig. S4)

line 303: "more similar". I guess the authors mean much more similar. RMS values would help.

line 339ff: check grammar

line 341 ff: I find it odd to focus so much on the RxxD motif. Rather, the authors have identified a ExR(X)6R motif in the N-domain (Fig. 8). (Please highlight also is Fig. S3). The representation of the homologs with putative c-di-GMP affinity should be improved. Would be nice to see a weblogo of the aligned sequences. Do they all have also the c-di-GMP binding residues of the C-domain conserved? How similar are these homologs? If they are all close homologs this analysis does not provide further insight in terms of the distribution of glycogen-debranching enzymes that are controlled by c-di-GMP vs. not controlled.

line 390: "GlgX contains a RxxD sequence for c-di-GMP binding" Phrase more carefully, since D is not involved in direct binding.

line 393: What is the main-chain conformation? Not a β -turn?

line 396: "it does not interact with a guanine base" I thought it does a stacking interaction?

line 405: "The strict conservation of key c-di-GMP binding residues ..." Show all residues that bind directly in Fig. S6.^[1]_{SEP}

We would like to thank the reviewer's for their helpful comments, critiques and suggestions as they have led to an improved manuscript. Below we provide a point-by-point response (in red).

Reviewer #1 (Remarks to the Author):

The authors present work of high significance and impact for the field by studying the function of cyclic di-GMP. c-di-GMP fulfills a global regulatory role in diverse bacterial functions, including biofilm formation, motility, virulence and cell cycle progression by binding to diverse protein effectors and riboswitches. In this manuscript the authors elucidate a novel allosteric regulatory function of the second messenger by characterizing its interaction with GlgX, the glycogen debranching enzyme in *S. venezuelae*. The authors verify c-d-GMP binding to Glgx, identify the binding site and later characterize a new binding motif. They establish c-d-GMP-mediated stimulation of GlgX activity and identify active site residues, which when mutated ablate GlgX activity. Beside a relatively weak attempt at analyzing GlgX substrate specificity, the authors demonstrate a key role of GlgX in *S. venezuelae* glycogen homeostasis by comparing strains expressing WT-levels of GlgX with those lacking or overexpressing GlgX. Finally, they solve multiple structures of GlgX and find support for c-di-GMP-mediated conformational change from inactive to active form of the enzyme. While the scientific approach and experimental seems generally appropriate, the work at times lacks in thoroughness necessary to make solid conclusions. Additional data is necessary to bolster some conclusions. The authors should include more details in the method section to improve the reproducibility of their work. Detailed concerns are found below.

We thank the reviewer for their careful review of our manuscript and excellent suggestions.

Major concerns:

Section starting line 120: It is not described how the stages vegetative growth (8 to 10 h), transition to sporulation (12 to 16 h) and sporulation (18 to 20 h) are confirmed? How do the authors know in which stage the culture is? To improve reproducibility it should be explained in more detail what time point the authors designate as time point zero. The conclusion of a 'rise of c-di-GMP during spore formation' is only valid if the authors confirm that the bacteria are in the assumed stages. In the Methods part should explain how the liquid culture of *S. venezuelae* is maintained, how cells are synchronized, when is "the start of the culture", and what are the conditions at this time point (cell density etc.)? In the results it should be mentioned how cell density changes over the experimental growth period.

We analyzed the samples harvested for nucleotide extraction using phase contrast microscopy and measured the optical densities of the cultures at 578 nm throughout the experiment. Representative images are now shown in Fig. S1. Following the suggestion of the reviewer, we have included the detailed data in the Methods. Our study takes advantage of *S. venezuelae*, which unlike *S. coelicolor*, sporulates in liquid culture, facilitating application of global techniques to the analysis of differentiation. However, currently, there are no means to synchronize the cells.

Section starting line 144: The authors generate two transgenic lines with two GlgX alleles (Vnz25270 and Vnz29170) expressed under the native promotor. Protein presence is tested by Flag Western-blot. The absence of expression of one of the Flag-GlgX proteins is interpreted as absence of the native form of this GlgX protein in the bacteria. However, this is only true under the assumption that the generation of both transgenic lines was equally successful. The authors should show mRNA expression levels of the transgenic/flag-tagged as well as of the WT alleles.

We re-evaluated our p3xFLAG::vnx25270 plasmid and *S. venezuelae* expressing vnx25270::FLAG. In the study by Miah et al, Microbiology, 2016, detection of vnx25270 transcripts using microarray transcriptional profiling has been reported. While these data show mRNA for vnx25270, they did not analyze protein levels. Again, we do not detect the Vnz25270::FLAG protein under the tested conditions. Our point in analyzing this protein was that it was not pulled down in our c-di-GMP capture

assay and hence we tested, by using purified recombinant Vnz25270, for c-di-GMP binding and we found, that, Vnz25270 does not bind c-di-GMP as does Vnz29170 (GlgX). Since Vnz25270 does not bind c-di-GMP, its expression is not a focus of this work, hence we have removed the former Fig. S1B and the associated text in the revised manuscript to focus on Vnz29170 (GlgX).

Figure 1C and Supplemental Figure S2: The authors compare binding of ligands to two purified proteins (Vnz25270 and Vnz29170) using DRaCALA assays. They draw conclusions about both proteins from an n of 1 experiment. The DRaCALA assays enable quantitative analyses. The authors should augment their data with appropriate technical replicates, and conduct quantitative analyses. The authors should also characterize the concentration and amount of radiolabel in the [32P]-c-di-GMP preparation they are using for their assay. The DRaCALA assays displayed in Figure S2 look unusual as the common outer circle is not visible, indicating changes in ligand concentration or radiolabel. To draw conclusions about ligand binding to both proteins, the authors should show purity analyses of both proteins. The authors do spot equal amounts of protein, but if the in the one case purification was not as effective, less of the GlgX paralogue will be spotted on the membrane, resulting in much less ligand retention in the center. Also, the authors should indicate which spots where developed imaged on the same membrane (for both figures) and can therefore be directly compared.

Thank you very much for this comment. In the revised version of the manuscript, we applied five different techniques to analyze c-di-GMP binding to GlgX: DRaCALA, nanoDSF assays, fluorescence polarization (FP), structural analysis and in the revision we added new data from microscale thermophoresis (MST). All these data, obtained by using different methods, support that c-di-GMP binds to GlgX. Two of the methods (FP and MST) allow robust quantitative analysis. Both FP and MST reveal that c-di-GMP binds to GlgX with an affinity of about 8 μ M. We apologize for having left out the concentration of radiolabeled c-di-GMP used in the DRaCALA assays. We have now added this information to the Methods section.

The outer ring in the DRaCALA data shown in former Fig. S2 (now Fig. S4), becomes visible when we increase the contrast settings (see Figure A below). We have now added to the figure legend and Methods section that both spots were analysed and imaged on the same membrane. The purity of the two proteins was checked using SDS PAGE and Coomassie staining. A representative image is included here. 2 μ g of each protein were loaded onto a 12.5 % PAA gel (see Figure B below).

Section starting line 179: The authors show maximal activation of GlgX by c-di-GMP at 50uM. However, they do not show that their reducing ends assay still responds to increased amounts of reducing ends. The assay should be run with D-glucose as a calibration standard for reducing ends and data should be expressed as ‘Reducing ends [xmol glucose equivalents]’.

In addition the stimulation should be presented in a scatter plot to better visualize the dose-response-relationship.

Thank you very much for this excellent suggestion. A calibration standard with D-glucose was run in parallel to the experiments. We have now included the standard curve in the new Fig. S7 and converted the OD562 values to ‘Reducing ends [μ M glucose equivalents]’ in all panels shown in Fig. 2. We included individual points in the bar graphs for the visualization of the data.

Regarding Fig. 2B: The authors should provide details regarding the origin of the glycogen, pullulan, amylopectin (vendor, catalogue number). The comparison of GlgX action on different carbohydrate substrates seems to be skewed by the use of much less carbohydrate starting material (significantly smaller signal at TLC origin for amylopectin and amylose). The authors should note that commercial preparations of polyglucans are of very different purity, equal carbohydrate concentration should therefore be based on glucose residues, not just based on weight dry material per volume. The TLC assay should be complemented with reducing end quantification, ideally including a quantification of branch-points per glucose unit for each carbohydrate substrate.

As requested, we have included the details of the sources for each carbohydrate in the Methods section. The reviewer raises an important issue concerning the purify of the carbohydrates. Upon our query, the technical service from Sigma-Aldrich informed us that for the carbohydrates used in our study, purity checks are not part of the quality control process. As suggested by the reviewer, we have performed BCA assays using glycogen, pullulan, amylopectin and amylose as substrates (new Fig. 2B). These data revealed that samples containing purified 1 μ M GlgX and 0.05% of each of the carbohydrate contain comparable, but not identical number of ‘Reducing ends [μ M glucose equivalents]’:

GlgX + Glycogen: 46.5 ± 4.5 ; increase at 50 μ M c-di-GMP to 164.5 ± 1.5 » 3.5 fold

GlgX + Pullulan: 48.7 ± 1.5 ; increase at 50 μ M c-di-GMP to 120.6 ± 10.6 » 2.5 fold

GlgX + Amylopectin: 29 ± 1.4 ; increase at 50 μ M c-di-GMP to 48.5 ± 3.4 » 1.7 fold

GlgX + Amylose: 29.1 ± 3.5 ; increase at 50 μ M c-di-GMP to 30.3 ± 3.3 » no increase

Moreover, this data set demonstrates that GlgX can generate reducing ends in amylopectin upon addition of c-di-GMP. Therefore, we conclude that the BCA assays not only provide the advantage of quantitative analysis but are also more sensitive than TLC analysis so that we have replaced the TLC data previously shown in Fig 2B with the BCA data. Taking into account the potential impurity issue of the different carbohydrates, we have rephrased the relevant text in the manuscript.

Line 196-198: The authors compare debranching efficiency a highly branched substrate (glycogen) with an essentially unbranched substrate (amylose) on the grounds of the amount of released chains. In the case of amylose the reaction is much less likely to happen in the first place, simply because they are just much lower in abundance. Why do the authors deduce lower GlgX efficiency from a lesser release of chains from amylose?

Thank you very much for pointing out this issue. As indicated above, the basal level of reducing ends in the reaction sample containing either amylopectin or amylose is almost identical. Still, we do detect GlgX-mediated degradation of amylopectin in presence of c-di-GMP but no increase of reducing ends can be detected when amylose is used as a substrate.

Thin layer chromatography details (e.g. plate type, spotting volume) are missing in the method section. The authors should explain what they think the signals away from the TLC origin signify in Fig. 2B. I assume it is glucan chains released by GlgX, with different mobility for different chain lengths. Should not debranching of pullulan lead to chain lengths of multiples of three (maltotriose * n)? However, the pattern of the released chains is identical to that released from glycogen. The authors have to explain that at least in the method part. Also, is it possible that chains were released from amylopectin, but they were too immobile to separate from the origin because as the authors state the amylopectin chains are much longer than those in glycogen?

Our new quantitative BCA analysis of GlgX activity using different carbohydrates shows that the BCA assay is more sensitive. Therefore, we replaced the TLC data with the new BCA data set (new Fig. 2B).

The authors should change all their bar graphs to include individual data points and indicate the error bar definition in each figure legend.

As requested by the reviewer, we have modified all bar graphs (Figs. 1A; 2A, 2B, 2C, 3B and Fig. S13E) and added the error bar definition to the relevant figure legends.

The effects of glgX knockout and overexpression on *Streptomyces* development could have an effect on glycogen levels. The authors state they did not see an effect of glgX deletion on development. They should explain and show the evidence for that. Also, the authors deduce from the absence of green spore pigment formation in glgX overexpressing cells that sporulation is delayed. To make this statement the authors must show time-dependent onset of sporulation. If they do not show that sporulation in overexpressing cells is still happening (but later than in WT) this is only speculation.

Streptomyces spore pigments are aromatic polyketides that are produced by enzymes encoded in the highly conserved *whiE* gene cluster. The expression of *whiE* genes is developmentally regulated and the synthesis of the spore pigment hence is representative of the last events in spore maturation (Kelemen et al., J. Bacteriol., 1998; Bush et al., NRM, 2015). As shown in Fig. 3C, the *glgX* mutant and the wild type are phenotypically indistinguishable and develop the green spore pigment to similar levels, while the strain overexpressing *glgX* remains white after the indicated time point. This allows the conclusion that spore maturation is affected upon *glgX* overexpression. We have now added a more detailed explanation to the text. Moreover, we have quantified the spores formed after 48 h of growth and confirmed that a strain overexpressing *glgX* produces a reduced number of viable spores when compared to wild type carrying the empty vector only. These data have now been added (new Fig. S6).

Figs 4 and 5: There is no reference in the text to figure 4B. 5A and 5C are referred to after introducing fig.6, which makes the whole thing a bit confusing. 5C could be moved to figure 1 and discussed earlier. There is a reference to 5B (line 236), but it doesn't seem to be showing what is referred to in the text. In line 239 there is a reference to fig 5D, which does not exist. The description for Figure 5B is missing in the figure legend. The authors should consolidate these things.

We thank the reviewer very much for pointing this out. The reference to 5B should have been 4B; The incorrect figure call outs have been fixed. The figure labels have also now been fixed. We added the figure legend for Fig. 5B. Again, we greatly appreciate the reviewer catching this and apologize for these issues.

The crosslinking experiment in Fig. S5 is not explained in sufficient detail. Was this done under physiological protein concentrations? How was this ensured? How did the authors verify that high MW bands are multimers of glgX? The lower DETR mutant gel shows much lower band intensities,

which indicates lower sensitivity of the signal detection. This could lead to an underestimation of the high MW bands. When I adjust the gels to similar intensity of the low MW bands (see figure below), it becomes apparent that crosslinking is very similar in the mutant protein, however not d-c-GMP dependent but only DSS dependent. The authors should discuss this. The authors use different size markers in both gels. Can the authors please indicate the commercial source?
[in the attached word doc there is an image displaying the Bottom gel of Fig. S3 with adjusted contrast

Additional details on the crosslinking experiment were now added to the Methods section and to the figure legend. 2 μ g of the proteins were used for crosslinking. Please note, that the physiological concentration of GlgX in *S. venezuelae* is unknown. We have now quantified the oligomeric forms of His₆-GlgX and His₆-GlgX_{DETR(49-52)} and added a new Fig. S9B to the supplement. These data confirm that the fraction of His₆-GlgX oligomers increased with increasing concentration of c-di-GMP, while the oligomeric fraction of His₆-GlgX_{DETR(49-52)} remained unaffected by c-di-GMP, consistent with our data showing that His₆-GlgX_{DETR(49-52)} does not bind c-di-GMP.

Please note that we have modified the gel images in the Fig. S9 to include the entire gel pictures. These images show that the His₆-GlgX and His₆-GlgX_{DETR(49-52)} proteins are pure. As such, the oligomeric bands that arise upon addition of DSS / c-di-GMP represent the His₆-GlgX and His₆-GlgX_{DETR(49-52)} proteins, respectively. The protein standards used in the assay are provided by NEB. Unfortunately, in the course of the experiments, the standard offered under the cat. no. P7712L was replaced by P7719L. We have now added the source for the standards to the figure legend.

Minor concerns:

Line 48: Since there is evidence that mammalian glycogen contains non-negligible amounts of glucosamine residues, the sentence should be changed to ‘Glycogen is a polysaccharide largely consisting of α -1,4-linked glucose subunits with α -1,6-linked glucose at the branching points.’
Changed as recommended.

Line 50: It should be made clear that only liver glycogen (not skeletal muscle glycogen) is considered to be used to replenish the blood with glucose. Nevertheless the muscle is playing a role in maintaining blood glucose levels by insulin-dependent glucose uptake and glycogen synthesis stimulation. There is a 2021 chapter on mammalian glycogen metabolism from Bart Pederson which summarizes the roles of glycogen in different organs quite nicely.

We have modified the sentence as suggested and added the reference.

Line 56: Not sure it makes sense to say “prolonged survival rates”. The survival is longer, the rates are higher... Are there such things as longer rates?

We changed to “prolonged survival”.

Line 68: Since this manuscript focuses on GlgX, it would be nice to characterize the debranching reaction bit more in detail. E.g. is the bacterial debranching reaction considered to be direct or indirect?

We have added to the introduction that bacterial glycogen debranching enzymes directly catalyze the hydrolysis of α -(1 \rightarrow 6)-linked glucose residues.

Line 147: The authors should include a reference to fig. S3 here.

We have included a reference to Fig S3 at this location. However, that changes the order of supplemental figure call out. This figure is now Fig. S2.

Line 156: The authors should clarify that they refer to Vnz29170 GlgX protein when they first speak about His6-tagged protein for nanoDSF and DRaCALA assays.

Done.

Section starting line 168: The current data doesn't rule out that c-di-GMP is still binding but not stabilizing the protein anymore. It is possible that the motifs are not necessary for binding but for stabilization. The authors should include the mutated GlgX proteins in their DraCALA analyses. Regarding Figure S4, the original RxxD motifs seem to be misspelled (i.e. RETR, and RLVR). Later in the manuscript the authors describe fluorescence polarization binding experiments that verify the binding data in Fig. 1. I suggest to move the FP data into figure 1 to substantiate the binding data when binding is discussed at first.

We thank the reviewer for pointing this out, indeed the RETR and RLVR in the figure legend should be RETD and RLVD. These have been fixed. There is a reason that the FP experiments were done at this later point and that was because without the structure it was not clear that the F-c-di-GMP molecule could be used to analyze binding. This is because the label is attached to one of the hydroxyl atoms of a ribose and without the structure it was not known whether a ribose would be exposed when in complex with GlgX, to allow such a molecule to be used. We have made this clearer in the revision. We performed a DRaCALA assay on the GlgX_{DETR(49-52)} variant (see below). Consistent with our thermal shift assay data (Fig. S5A), BCA assay data (Fig. 2C), and structural analysis identifying the c-di-GMP binding signature (Fig. 5), this assay showed that the GlgX_{DETR(49-52)} variant does not bind c-di-GMP (these data were added to Fig. S5).

Line 190: Amylose should be called 'essentially unbranched chain'. The amylopectin description seems a bit peculiar. The authors should add a citation to identify the source of their information.

We have changed the description of amylose to essentially unbranched chain and modified the definition of amylopectin according to Carvalho A.J.F, Handbook of Biopolymers and Biodegradable Plastics, 2013.

Fig. S3: The authors should indicate in the figure legend what '.*' symbols signify.

This information has been added.

Reviewer #2 (Remarks to the Author):

In this recent work the Tschowri/Schumacher team uncovers the regulation and molecular mechanism of glycogen storage control by the second messenger c-di-GMP in *Streptomyces*, an organism that is a major source of antibiotics involving a complex developmental program. Although c-di-GMP has been linked to several pathways pertaining to biofilm formation, motility, virulence and cell cycle regulation, this report is the first in-depth study into the second messenger's role in regulating glycogen metabolism, representing a truly novel finding that should be of great interest to

the field. Previously c-di-GMP has been established as a major regulator for the transition from a vegetative to spore forming state of *Streptomyces* due to its ability to bind to and regulate effector proteins BldD and WhiG, central regulators for distinct steps in the pathway. Cellular c-di-GMP levels are high during the vegetative growth phase but drop as the developmental transition proceeds. Here, the authors show that there is a second phase where c-di-GMP levels rise, namely as cells sporulate. Using an unbiased affinity enrichment for c-di-GMP-binding proteins, the authors identify the glycogen-debranching enzyme GlgX as a specific interactor for the second messenger. GlgX enzyme activity is stimulated in the presence of c-di-GMP and the enzyme is crucial for glycogen breakdown in *Streptomyces*. Crystal structures of *Streptomyces* GlgX in its apo-form, bound to c-di-GMP and bound to c-di-GMP and a substrate analog reveal the molecular mechanism regarding the activation of GlgX by the dinucleotide. Specifically, the apo-form is monomeric and displays an active site in a catalytically incompetent state. C-di-GMP binding, via a long-range, allosteric effect, reorders the active site, with the enzyme adopting an active state that can accommodate substrate. The active state appears to involve a weak GlgX dimer, although the functional relevance of this detail was not thoroughly tested here. Notably, the specific interaction of c-di-GMP with GlgX expands the repertoire of reported cyclic dinucleotide binding sites. The experiments presented in the manuscript contain important controls, including sequence- and structure-informed mutations, linking the structural insight to the regulation of GlgX in cells. The flow from identifying a second phase of c-di-GMP during *Streptomyces* spore formation to the identification of a crucial metabolic enzyme to its structure in multiple functional snapshots depicting the mode of c-di-GMP activation of GlgX renders this report complete with several important, novel discoveries on the way.

We thank the reviewer for their positive comments and very helpful suggestions and critiques.

The only major point that I feel was not fully addressed pertains to the dimerization of GlgX and its relevance for the activation of the enzyme by c-di-GMP: The crosslinking data in Figure S5 are not fully convincing in showing dimerization of GlgX in response to c-di-GMP. This is in line with a modest, yet potentially sufficient dimerization interface observed in crystals, but its impact on the enzymatic mechanism was not tested. Have the authors tried other methods for reading out oligomerization? Gel filtration (ideally coupled with a multi-light scattering setup), analytical ultracentrifugation or mass photometry could yield more quantitative data. Along the same lines: While the authors have done a great job validating the structural insight by using site-directed mutants designed based on the structural models, such an analysis was omitted for the dimerization interface. This leaves open a mechanistic question: Is dimerization required for GlgX activity? Would it be possible to design a mutant variant of GlgX that would be predicted to still bind c-di-GMP and substrate but that would be impaired in dimerization? Or would a dimerization mutant be defective in c-di-GMP and/or substrate binding?

The reviewer's point is well taken. Addressing the relevance of dimerization to enzyme activity is challenging because each end of the dimer is involved in c-di-GMP binding and the interaction of c-di-GMP with the C-domain is crucial for activity due to the induced conformational change to the active site. Thus, to address this question of dimerization and its possible involvement in activity, we have 1) quantified crosslinking experiments done in the presence and absence of c-di-GMP, which were performed at relatively low protein concentrations (2 μ g). The results show that more dimer is formed in the presence of c-di-GMP (Fig. S9). 2) We also performed new SEC analyses of the apo and c-di-GMP bound GlgX (Fig. S10). However, these studies were carried out at relatively high protein concentrations, 4 mg/mL, and we see dimer in the presence and absence of c-di-GMP at these protein concentrations. This is also consistent with our crystal structures; the apo structure reveals the similar dimer as the c-di-GMP bound dimer, when looking at symmetry related molecules, indicating that the dimer can form in the apo state without c-di-GMP at high concentrations. 3) Hence, to really get at the central question of whether dimerization is important for GlgX activity (while still allowing

c-di-GMP binding and its concomitant conformational change through the C-domain), we made a GlgX fusion mutant protein (see details in text and new Fig. S13) in which we mutated residues at the N-subdomain to prevent c-di-GMP binding to this region as well as residues in the central dimer interface and then attached covalently via a long (GGGSGGGSGGGSGGGSGGG) linker a functional N-domain so that, importantly, c-di-GMP binding could be retained to signal through the C-domain. In addition, no residues were mutated in the active site. See below for the details of the design strategy. We note also that the inclusion of the N-domain at the C-terminus and its interaction with the C-domain would prevent the formation of a dimer from this construct. We were able to purify the construct and we then performed a battery of tests to assess if the protein was folded (circular dichroism), can bind c-di-GMP (using FP) and to determine its oligomeric state (SEC) (see Figure S13A-D). These studies showed that this GlgX fusion mutant is folded, can bind c-di-GMP and is monomeric. We then tested the enzymes activity. The enzyme assays showed that, indeed, the monomeric protein is not active even with added c-di-GMP. These data have now been added to the manuscript (and see Figure S13 for details of the design, CD, FP, SEC and enzyme activity assays).

GlgX fusion mutant design and characterization

Minor points:

1. The supplemental movie is effective in presenting the overall structure and the mode of c-di-GMP binding. Have the authors considered illustrating the conformational changes from the apo to the c-di-GMP-bound state? This could supplement the still images shown in the figures.

We prefer not to utilize morphing movies, which require the generation of several intermediate states that are not based on experimental data. We do not have any information on the steps from one state to the next (we only have the end states) and we wish not to speculate.

2. Figure 7: This figure shows a superposition of the c-di-GMP bound state with a substrate analog- and c-di-GMP-bound conformation. It would be informative to also show a close-up view of the substrate analog-bound active site with that of the apo-GlgX structure superimposed. This would highlight the structural changes upon activation by c-di-GMP relative to the substrate, adding clarity to the mechanism of activation.

As suggested, we have added this as a supplemental figure (new Fig. S12).

3. Have the authors attempted to crystallize a c-di-GMP-free GlgX bound to the substrate analog?

Along those lines, does substrate or substrate analog alter the oligomeric state of GlgX in the presence or absence of c-di-GMP?

Some setups were done of the protein with acarbose and no c-di-GMP but no crystals produced. However, only a few such screens were completed as when our data indicated the enzyme is not active without c-di-GMP we focused efforts on the GlgX-c-di-GMP-acarbose complex.

Reviewer #3 (Remarks to the Author):

Schumacher et al. report about the regulation of a glycogen-debranching enzyme (GlgX) from a *Streptomyces* species by the second messenger c-di-GMP. They show direct binding of c-di-GMP and saturable activation of the enzymatic activity by the compound. Several crystal structures are presented which reveal the c-di-GMP binding site, which is validated by mutagenesis, and the structural changes induced by c-di-GMP binding. The study has been performed competently and the results are presented well. GlpX is thus one of the few c-di-GMP targets that have been studied thoroughly both biochemically and structurally. However, it is expected that there are many more unrelated c-di-GMP binding proteins, such that the results of the current study cannot be generalized, in particular with respect to the mode of c-di-GMP binding and the induced allosteric change.

Therefore, I think that this paper would be more suited to be published in a specialized journal.

Below, I briefly list a few points that occurred to me upon first reading:

We thank the reviewer for their thorough reading and insightful comments.

* Why is c-di-GMP sensed only in the late developmental stage (l. 140)? This is mentioned again in the Discussion (l. 369). I guess it would be due to the higher K_d for GlpX binding (compared to the other c-di-GMP binders mentioned). Would be appropriate to elaborate on this. Also discuss the effect of high c-di-GMP levels on BldD, etc.

The interaction between GlgX and c-di-GMP is likely favored only during the late developmental stage for two reasons. First, the protein is particularly abundant during sporulation (Fig. S3). Second, GlgX has a relatively low affinity for c-di-GMP ($\sim 8 \mu\text{M}$; see Fig. 5C and S11) and therefore needs higher levels of c-di-GMP that are detectable in the late growth stages (Fig. 1A) for activation. We have added this information to the discussion.

BldD and WhiG-RsrG would respond to high [c-di-GMP] in the late stages of the life cycle, provided the proteins are present. However, using a related model, *Streptomyces coelicolor*, the study by den Hengst et al, Mol Microbiol, 2010, demonstrated that levels of BldD decrease in the late developmental stage, indicating that, due to protein degradation, BldD would not respond to c-di-GMP increase in the late stage. However, for the model used in this work, *S. venezuelae*, BldD protein levels have not been measured. In addition, no data are available on the activity of BldD, e.g. expression of target genes that solely depend on BldD in *S. venezuelae*. In the case of WhiG, the protein appears to be stable throughout the growth, but seems to become inactive in the late stages of development, as judged by the mRNA levels of WhiG target genes (Gallagher and Schumacher et al., Mol Cell, 2020). Due to the incomplete data on BldD and WhiG in *S. venezuelae*, we did not include this in our discussion as it would be highly speculative, and the focus of the work is GlgX.

* Binding of c-di-GMP to GlpX has been studied by several techniques (paragraph starting at l. 138), but the quantitative analysis by fluorescence polarisation measurement is reported only much later (l. 270ff). Latter study employed FLUO-c-di-GMP ($K_d=8 \mu\text{M}$). To get the affinity of unmodified c-di-GMP a corresponding competition experiment is missing.

We have now included the results of c-di-GMP binding to GlgX using another method (MST) that does not have the fluorescent tag. See Fig. S11. There is a reason that the FP experiments were done at this later point and that was because without the structure it was not clear that the F-c-di-GMP molecule could be used to analyze binding, because the label is attached to one of the hydroxyl atoms

of a ribose. Without the structure it was unclear that a ribose would be exposed to allow such a molecule to be used. We have made this clearer in the revision.

line 29: GlgX-c-di-GMP and apo GlgX structures reveal >> comparison of the GlgX-c-di-GMP and apo GlgX structures reveals

We have changed the sentence as suggested by the reviewer.

line 176: “that the RxxD motif at position 49-52 is essential for ligand binding”. The conclusion reaches too far. The swap experiment reveals that the R, the D, or both are essential for binding.

We have revised this sentence.

line 186: “maximal enzymatic activity in the presence of 50 μ M c-di-GMP” >> activity is saturating at 50 μ M c-di-GMP. By proper quantitative evaluation of the activation curve (probably requiring further experimental data) it should be possible to get the activation constant or EC50 of the c-di-GMP activator. Would be interesting to compare this with the binding data.

Thank you very much, the text was changed accordingly. Following the reviewer’s suggestion, we have calculated the half maximal effective concentration (EC50) of c-di-GMP in GraphPad Prism 9 using the [Agonist] vs. response - Variable slope (four parameters) equation within the Nonlinear regression (curve fit) analysis (see Figure below). The calculated EC50 value is 16.52 μ M. The affinity of GlgX for c-di-GMP has been determined to be about 8 μ M using two different methods, fluorescence polarisation (Fig. 5C) and microscale thermophoresis (Fig. S11). While the affinity and the EC50 value are in a comparable range, the 2-fold difference may result from different conditions e.g. buffer composition in the assays.

line 227: mention phasing method

We noted that the apo GlgX structure was solved by Molecular replacement using the TreX (2VUY) structure as a search model. Subsequent GlgX structures were solved using our *S. venezuelae* GlgX structure and rebuilt as necessary. We have added more details to the revision.

line 236: Fig. 5B >> Fig. 4B

Done

line 236: change no >>> There are no protein-protein interfaces in the crystal that bury a significant

amount of buried surface area

This section was rewritten.

line 267: “showing that the RETD49-52 motif is essential for c-di-GMP binding. “ Up to here only the involvement of R49 has been shown.^[1]_[SEP]

We have reworded this sentence.

line 277: Our analyses showed >> As shown above (Fig. S4)

changed, as suggested.

line 303: “more similar”. I guess the authors mean much more similar. RMS values would help.

We have added this information.

line 339ff: check grammar

We thank the reviewer for catching this, which has been fixed.

line 341 ff: I find it odd to focus so much on the RxxD motif. Rather, the authors have identified a ExR(X)6R motif in the N-domain (Fig. 8). (Please highlight also is Fig. S3). The representation of the homologs with putative c-di-GMP affinity should be improved. Would be nice to see a weblogo of the aligned sequences. Do they all have also the c-di-GMP binding residues of the C-domain conserved? How similar are these homologs? If they are all close homologs this analysis does not provide further insight in terms of the distribution of glycogen-debranching enzymes that are controlled by c-di-GMP vs. not controlled.

We have revised the discussion to focus on the finding of this new c-di-GMP motif. We, however, mention the presence of the RxxD as it was discovered in our studies, prior to obtaining structures, as being important in c-di-GMP binding. As we detail, while the Arginine in the motif is involved in binding to c-di-GMP, the Aspartic acid is not. Moreover, most RxxD motifs bind a intercalated c-di-GMP dimer. The residues that mediate specific binding to c-di-GMP as revealed in our structure, make up the ExR(x)6R motif. We have included the c-di-GMP binding residues of the C-domain in Fig. S3 (new Fig. S2). Moreover, as suggested by the reviewer, we have now generated a WebLogo based on an alignment between GlgX homologues that are listed in Table S3. For better visualization and following the reviewer’s suggestion, we now highlight the c-di-GMP binding residues located in the N- and C-terminus of GlgX in the WebLogo (new Fig. S14), instead of the sequences in Table S3. In the revised version of Table S3, we have included a “percent identity” column, showing that GlgX homologs containing the c-di-GMP binding signature contain 43-97% identical residues. Thus, GlgX homologs show sequential differences, yet they display a high degree of conservation of the c-di-GMP binding signature residues, in line with our conclusion that control of GlgX activity by c-di-GMP is highly conserved within Actinobacteria.

line 390: “GlgX contains a RxxD sequence for c-di-GMP binding” Phrase more carefully, since D is not involved in direct binding.

We have rewritten this sentence to be more specific.

line 393: What is the main-chain conformation? Not a β -turn?

The residues are in a loop. This has been added.

line 396: “it does not interact with a guanine base” I thought it does a stacking interaction?

The numbering was off here, and the reviewer is correct, the arginine makes stacking interactions (which has now been added). However, unlike the R from other RxxD motifs, it does not make

specifying hydrogen bonds to a guanine base. We thank the reviewer for pointing this out.

line 405: “The strict conservation of key c-di-GMP binding residues ...” Show all residues that bind directly in Fig. S6.^[1]

Following the reviewer’s suggestion above, we now show a WebLogo in which we highlighted the residues involved in c-di-GMP binding (new Fig. S14).

Reviewer #1 (Remarks to the Author):

The authors revised and significantly improved their present work, which is of high significance and impact for the field as it studies the function of cyclic di-GMP. c-di-GMP fulfills a global regulatory role in diverse bacterial processes, including biofilm formation, motility, virulence and cell cycle progression by binding to diverse protein effectors and riboswitches. In this manuscript the authors elucidate a novel allosteric regulatory function of the second messenger by characterizing its interaction with GlgX, the glycogen debranching enzyme in *S. venezuelae*. The authors verify c-di-GMP binding to GlgX, identify the binding site and later characterize a new binding motif. They establish c-di-GMP-mediated stimulation of GlgX activity and identify active site residues, which when mutated ablate GlgX activity.

While the majority of concerns were appropriately addressed, the point of c-di-GMP-dependent dimerization (and hence activation) of GlgX is still not convincing. (1) c-di-GMP dependence is based largely on the n of one experiment in Fig. S9. The authors should include repeats of the experiment in Fig. S9 and plot the average of several experiments, to make the case that GlgX dimerization is indeed c-di-GMP-dependent. (2) The authors expressed an GlgX fusion mutant protein, which was unable to dimerize yet still binding c-di-GMP. Conceptually it is hard to prove that the inability of this GlgX-mutant to form dimers ablates the enzymatic activity, or whether the experimental changes in the protein led to other yet undetected structural deficits that simply disabled the enzyme. Eventual prove could come from implementing a molecular switch to experimentally control homo-dimerization (for instance reviewed here: *Front Chem.* 2022; 10: 829312), where dimerization (and activity?) can be restored. In the results the authors argue that dimerization of GlgX leads to activation, but they fail to sufficiently discuss the caveats of their approach.

In addition, the saturating concentration of c-di-GMP (Line 189) is still not proven. The authors added a calibration curve. However, this curve ends at 100uM glucose. During glycogen debranching at 50uM c-di-GMP, the reducing end concentration generated already exceeds the range of the standard curve (about 150uM glucose equivalents). It is unclear whether the BCA assay still responds to higher concentrations of reducing ends, such as when 100uM c-di-GMP are applied. Please include a standard curve that includes at least 200uM glucose equivalents or remove the 100uM c-di-GMP bar as well as the statement about the saturating c-di-GMP concentration.

Line 468: 'scrapped', change to 'scraped'

Reviewer #2 (Remarks to the Author):

The authors have addressed all my points in great detail, including some elegant experiments to probe the activity of a monomeric enzyme. I have no further comments and congratulate the authors to this interesting work.

Reviewer #3 (Remarks to the Author):

The authors have improved the work considerably by doing additional experiments and changes to the manuscript. All my remarks and suggestions have been answered satisfactorily.

NB. I can't find Table S3.

1 We thank the reviewers for their in-depth reviews. Below we provide a point-by-point response to
2 the remaining comments from the reviewers (our responses in red).

3 4 **RESPONSE TO REVIEWER COMMENTS**

5 **Reviewer #1 (Remarks to the Author):**

6 The authors revised and significantly improved their present work, which is of high significance
7 and impact for the field as it studies the function of cyclic di-GMP. c-di-GMP fulfills a global
8 regulatory role in diverse bacterial processes, including biofilm formation, motility, virulence and
9 cell cycle progression by binding to diverse protein effectors and riboswitches. In this manuscript
10 the authors elucidate a novel allosteric regulatory function of the second messenger by
11 characterizing its interaction with GlgX, the glycogen debranching enzyme in *S. venezuelae*. The
12 authors verify c-d-GMP binding to Glgx, identify the binding site and later characterize a new
13 binding motif. They establish c-d-GMP-mediated stimulation of GlgX activity and identify active
14 site residues, which when mutated ablate GlgX activity.

15 We would like to thank the reviewer for all his/her very helpful comments and critiques.

16
17 While the majority of concerns were appropriately addressed, the point of c-di-GMP-dependent
18 dimerization (and hence activation) of Glgx is still not convincing. (1) c-di-GMP dependence is
19 based largely on the n of one experiment in Fig. S9. The authors should include repeats of the
20 experiment in Fig. S9 and plot the average of several experiments, to make the case that Glgx
21 dimerization is indeed c-di-GMP-dependent.

22 We need to clarify that we do not indicate that our data shows GlgX dimerization is dependent on c-
23 di-GMP, but rather that c-di-GMP facilitates dimerization. Specifically, on page 11 we say that the
24 data indicates that “GlgX likely exists in a monomer to dimer equilibrium, with c-di-GMP
25 appearing to stabilize the dimer”. We have now added more text to clarify this in the revision.

26 Aside from dimerization, a central point is that c-di-GMP binding to GlgX is essential for its
27 activity because its binding induces long range structural changes that are transmitted to the active
28 site, leading to an active conformation, independent from its effect on dimerization. Again, we have
29 added discussion to try and ensure these points are clarified in the revision,

30
31 (2) The authors expressed an Glgx fusion mutant protein, which was unable to dimerize yet still
32 binding c-di-GMP. Conceptually it is hard to prove that the inability of this Glg-mutant to form
33 dimers ablates the enzymatic activity, or whether the experimental changes in the protein led to
34 other yet undetected structural deficits that simply disabled the enzyme. Eventual prove could come
35 from implementing a molecular switch to experimentally control homo-dimerization (for instance
36 reviewed here: Front Chem. 2022; 10: 829312), where dimerization (and activity?) can be
37 restored. In the results the authors argue that dimerization of Glgx leads to activation, but they fail
38 to sufficiently discuss the caveats of their approach.

39 This is an excellent point. We agree with the reviewer that there could be undetected structural
40 changes due to the fusion mutant construction, even though we showed by CD that it is in a folded
41 state. We have now added this important caveat to the manuscript. Specifically, we indicated in the
42 revised text that we cannot exclude the possibility that the engineered mutant protein has some
43 undetected change(s) in its structure that could impact activity. The utilization of a molecular
44 switch is an interesting idea. Utilization of a molecular switch, glue-based approach to restore
45 activity of the GlgX fusion mutant, if that is the reviewer’s suggestion, would not be applicable here
46 because the mutations in the central domain were designed to prevent the specific dimer interaction.
47 Utilization of the fusion mutant was the only way we could envision to test the issue of dimer
48 activity. But as noted, we have now included the caveat to this approach in the revision. We thank
49 the reviewer for the suggestions.

50
51 In addition, the saturating concentration of c-di-GMP (Line 189) is still not proven. The authors
52 added a calibration curve. However, this curve ends at 100uM glucose. During glycogen

53 debranching at 50uM c-di-GMP, the reducing end concentration generated already exceeds the
54 range of the standard curve (about 150uM glucose equivalents). It is unclear whether the BCA
55 assay still responds to higher concentrations of reducing ends, such as when 100uM c-di-GMP are
56 applied. Please include a standard curve that includes at least 200uM glucose equivalents or remove
57 the 100uM c-di-GMP bar as well as the statement about the saturating c-di-GMP concentration. **We**
58 **apologize for not having addressed the issue raised by the reviewer to full satisfaction during the**
59 **first revision of the manuscript. As requested by the reviewer we performed the calibration (see**
60 **below), adding points at 120 uM, 140 uM, 160 uM, 180 uM and 200 uM. As shown in the figure**
61 **below, the assay reaches saturation at 100 μM glucose. For the enzyme assays shown in Fig. 2 and**
62 **the standard curve in Fig. S7, we used the identical stock solution of reagents (250 mM Na₂CO₃;**
63 **144 mM NaHCO₃; 2.5 mM sodium bicinconinic acid; 6 mM L-serine; 2.5 mM CuSO₄ x 5 H₂O).**
64 **Therefore, we retain the standard curve currently shown in Fig. S7.**
65

80

81 Line 468: 'scrapped', change to 'scraped'

82 **Changed as suggested. Thanks to the reviewer for catching this.**

83

84 **Reviewer #2 (Remarks to the Author):**

85 The authors have addressed all my points in great detail, including some elegant experiments to
86 probe the activity of a monomeric enzyme. I have no further comments and congratulate the authors
87 to this interesting work.

88 **We thank the reviewer again for all his/her very helpful comments and suggestions.**

89

90 **Reviewer #3 (Remarks to the Author):**

91 The authors have improved the work considerably by doing additional experiments and changes to
92 the manuscript . All my remarks and suggestions have been answered satisfactorily.

93 **We thank the reviewer for their insightful comments and critiques.**

94

95 NB. I can't find Table S3.

96 **We appreciate the reviewer for catching this. This table was mistakenly labeled as a supplemental**
97 **figure and this has now been fixed.**

98

No Reviewer Comments:

1 RESPONSE TO REVIEWER COMMENTS

2 Reviewer #1 (Remarks to the Author):

3 The authors revised and significantly improved their present work, which is of high significance
4 and impact for the field as it studies the function of cyclic di-GMP. c-di-GMP fulfills a global
5 regulatory role in diverse bacterial processes, including biofilm formation, motility, virulence and
6 cell cycle progression by binding to diverse protein effectors and riboswitches. In this manuscript
7 the authors elucidate a novel allosteric regulatory function of the second messenger by
8 characterizing its interaction with GlgX, the glycogen debranching enzyme in *S. venezuelae*. The
9 authors verify c-d-GMP binding to Glgx, identify the binding site and later characterize a new
10 binding motif. They establish c-d-GMP-mediated stimulation of GlgX activity and identify active
11 site residues, which when mutated ablate GlgX activity.

12 **We would like to thank the reviewer for all his/her very helpful comments and critiques.**

13
14 While the majority of concerns were appropriately addressed, the point of c-di-GMP-dependent
15 dimerization (and hence activation) of Glgx is still not convincing. (1) c-di-GMP dependence is
16 based largely on the n of one experiment in Fig. S9. The authors should include repeats of the
17 experiment in Fig. S9 and plot the average of several experiments, to make the case that Glgx
18 dimerization is indeed c-di-GMP-dependent.

19 (2) The authors expressed an Glgx fusion mutant protein, which was unable to dimerize yet still
20 binding c-di-GMP. Conceptually it is hard to prove that the inability of this Glg-mutant to form
21 dimers ablates the enzymatic activity, or whether the experimental changes in the protein led to
22 other yet undetected structural deficits that simply disabled the enzyme. Eventual prove could come
23 from implementing a molecular switch to experimentally control homo-dimerization (for instance
24 reviewed here: Front Chem. 2022; 10: 829312), where dimerization (and activity?) can be
25 restored. In the results the authors argue that dimerization of Glgx leads to activation, but they fail
26 to sufficiently discuss the caveats of their approach.

27 **After discussing all the issues with the dimerization data with the editor, we decided to remove these**
28 **data (crosslinking and fusion mutant analyses) as there was agreement that they did not add**
29 **significantly to the main findings of the study, but rather detracted from the main points of the work.**
30 **The current revision is much more focused and importantly, serves to highlight the key findings of**
31 **the study.**

32
33 In addition, the saturating concentration of c-di-GMP (Line 189) is still not proven. The authors
34 added a calibration curve. However, this curve ends at 100uM glucose. During glycogen
35 debranching at 50uM c-di-GMP, the reducing end concentration generated already exceeds the
36 range of the standard curve (about 150uM glucose equivalents). It is unclear whether the BCA
37 assay still responds to higher concentrations of reducing ends, such as when 100uM c-di-GMP are
38 applied. Please include a standard curve that includes at least 200uM glucose equivalents or remove
39 the 100uM c-di-GMP bar as well as the statement about the saturating c-di-GMP concentration. **We**
40 **apologize for not having addressed the issue raised by the reviewer to full satisfaction during the**
41 **first revision of the manuscript. As requested by the reviewer we performed the calibration (see**
42 **below), adding points at 120 uM, 140 uM, 160 uM, 180 uM and 200 uM. As shown in the figure**
43 **below, the assay reaches saturation at 100 μM glucose. For the enzyme assays shown in Fig. 2 and**
44 **the standard curve, we used the identical stock solution of reagents (250 mM Na₂CO₃; 144 mM**
45 **NaHCO₃; 2.5 mM sodium bicinconinic acid; 6 mM L-serine; 2.5 mM CuSO₄ x 5 H₂O). Therefore,**
46 **we retain the standard curve shown in Fig. S10.**

47
48
49
50
51
52

53
54
55
56
57
58
59
60
61
62
63
64
65
66
67
68
69
70
71
72
73
74
75
76
77
78
79
80
81
82
83
84
85
86
87

Line 468: ‘scrapped’, change to ‘scraped’

Changed as suggested. Thanks to the reviewer for catching this.

Reviewer #2 (Remarks to the Author):

The authors have addressed all my points in great detail, including some elegant experiments to probe the activity of a monomeric enzyme. I have no further comments and congratulate the authors to this interesting work.

We thank the reviewer again for all his/her very helpful comments and suggestions.

Reviewer #3 (Remarks to the Author):

The authors have improved the work considerably by doing additional experiments and changes to the manuscript . All my remarks and suggestions have been answered satisfactorily.

We thank the reviewer for their insightful comments and critiques.

NB. I can't find Table S3.

This table was mistakenly labeled as a supplemental figure and has now been fixed (changed to supplemental Table).

No Reviewer Comments:

1 **Response to reviewers comments:**

2 As seen below, there were no further comments.

3

4 **REVIEWERS' COMMENTS**

5

6

7

8

9 ** See Nature Portfolio's author and referees' website at www.nature.com/authors for information about
10 policies, services and author benefits

11

12

13

14